# Evolutionary origin of Hoxc13-dependent skin appendages in amphibians

Marjolein Carron [1,2,5], Attila Placido Sachslehner [3,5], Munevver Burcu Cicekdal [1,2], Inge Bruggeman[1,4], Suzan Demuynck[1], Bahar Golabi[3], Elfride De Baere [2], Wim Declercq [1,4], Erwin Tschachler[3], Kris Vleminckx [1] ✉ & Leopold Eckhart [3] ✉

Cornified skin appendages, such as hair and nails, are major evolutionary innovations of terrestrial vertebrates. Human hair and nails consist largely of special intermediate filament proteins, known as hair keratins, which are expressed under the control of the transcription factor Hoxc13. Here, we show that the cornified claws of *Xenopus* frogs contain homologs of hair keratins and the genes encoding these keratins are flanked by promoters in which binding sites of Hoxc13 are conserved. Furthermore, these keratins and Hoxc13 are co-expressed in the claw-forming epithelium of frog toe tips. Upon deletion of *hoxc13*, the expression of hair keratin homologs is abolished and the development of cornified claws is abrogated in *X. tropicalis*. These results indicate that Hoxc13-dependent expression of hair keratin homologs evolved already in stem tetrapods, presumably as a mechanism for protecting toe tips, and that this ancestral genetic program was coopted to the growth of hair in mammals.

The evolution of terrestrial vertebrates is characterized by the appearance of an efficient barrier against water loss in a dry environment and by the evolution of hard cornified skin appendages, such as claws, scales, feathers, and hair, which are critical for the capture of prey, protection, support of special modes of locomotion, and thermoinsulation[1–3]. Cornified skin appendages are characteristic features of important phylogenetic clades of terrestrial vertebrates, exemplified by hair in mammals and feathers in birds. Thus, the evolution of skin appendages is a major research question in the biology of tetrapods[2].

Cornified skin appendages consist of terminally differentiated and inter-connected keratinocytes which are packed with specific cytoskeletal and other structural proteins. The keratinocytes within the mature parts of hard skin appendages such as hair, feathers, claws and nails are dead, meaning that they lack gene expression and metabolism. Importantly, the aforementioned skin appendages are regenerated either continuously in the case of cornified claws and nails or cyclically in the case of hair and feathers. Thus, the program of keratinocyte differentiation and cornification is not only active during embryonic development but also in the adult organism. Mammalian cornified skin appendages are characterized by the expression of so-called hair keratins, i.e. keratin intermediate filament proteins which were originally identified in hair[4]. Human hair keratins are comprised of KRT31-KRT40, which are type I keratins, and KRT81-KRT86, which are type II keratins[5]. Type I and type II hair keratins heterodimerize to form intermediate filaments. Not only hair, but also claws, nails, horns, filiform papillae of the tongue, and other hard skin appendages of mammals contain hair keratins[5]. We identified homologs of hair keratins in the green anole lizard and showed that these keratins are expressed in the cornified claws[6], which are homologous to human nails. The conserved expression in claws and nails suggested that (1) proteins of the type I and II "hair keratin" families originated prior to the divergence of mammals and reptiles, and (2) the primordial sites of "hair keratin expression" were not hair follicles but other structures that were likely homologous to claws in extant reptiles and mammals. Further studies led to the identification of hair keratin-like proteins, though distinguished from human hair keratins by a low content of

[1]Department of Biomedical Molecular Biology, Ghent University, 9000 Ghent, Belgium. [2]Department of Biomolecular Medicine, Ghent University and Center for Medical Genetics, Ghent University Hospital, 9000 Ghent, Belgium. [3]Department of Dermatology, Medical University of Vienna, 1090 Vienna, Austria. [4]VIB-Ugent Center for Inflammation Research, 9000 Ghent, Belgium. [5]These authors contributed equally: Marjolein Carron, Attila Placido Sachslehner. ✉e-mail: kris.vleminckx@irc.ugent.be; leopold.eckhart@meduniwien.ac.at

cysteine, in amphibians[7–9]. These homologs of hair keratins are expressed in toe pads of tree frogs[10], but it has remained unknown where they are expressed in phylogenetically diverse amphibians with claw-like structures, such as clawed frogs of the genus *Xenopus*, clawed salamanders of the genus *Onychodactylus* and some sirens such as *Pseudobranchus striatus*[11]. Because of developmental and morphological differences to claws of amniotes, such as the accumulation of parallel cornified cell layers in claws of *Xenopus* frogs as opposed to the proximal-to-distal growth of human and mouse nails and claws of lizards, it was proposed that claws of clawed frogs had evolved independently from claws of amniotes[11,12]. However, a conclusive molecular analysis of amphibian claws has not been reported yet.

While structural proteins as components of cornified keratinocytes determine the hardness of skin appendages, the spatial and temporal pattern of expression of the corresponding genes and thereby the development and continuous regeneration of hard skin appendages are regulated at the level of gene transcription. One of the best characterized regulators of mammalian hair keratins is the transcription factor Hoxc13, which is expressed in the hair matrix and nail matrix[13,14]. Hoxc13 induces the expression of several hair keratin genes by binding to specific sites in the proximal promoters of hair keratin genes[14]. Mutations in human *HOXC13* cause ectodermal dysplasia 9, a severe defect of hair and nails[15,16]. Likewise, mutations of *Hoxc13* in mice, rabbits and pigs suppressed the growth of hair and nails[17–19]. The primordial function of Hox genes is to specify regions of the body along the head-tail axis during embryonic development[20–22], whereby *Hoxc13* as the terminal gene of the HoxC gene cluster is activated in the posterior part of the body, such as the tail fin of Actinopterygii (ray-finned fishes)[23]. During the evolution of Sarcopterygii (lobe-finned fishes), *Hoxc13* was coopted to expression in paired fins, the evolutionary precursors of limbs in tetrapods[24]. Recent research has identified enhancer elements critical for the expression of HoxC genes in developing nails and hair follicles of mammals[25]. Importantly, the evolutionary steps between the role of Hoxc13 in fins of early Sarcopterygians and the regulatory role of Hoxc13 in mammalian hair and nails have remained elusive.

Here, we investigated the evolutionary origin of the link between Hoxc13 and hair keratins. We show that amphibian homologs of hair keratins are components of claws in the Western clawed frog (*X. tropicalis*), a model for basal tetrapods. The expression of these keratins and the formation of cornified claws were abrogated by the targeted disruption of *hoxc13* in *X. tropicalis*, which is equivalent to the absence of hair and nails in mammals upon genetic inactivation of *Hoxc13*. Our results highlight a conserved role of Hoxc13 in claws and hair and link the evolution of limbs with the evolution of cornified skin appendages in tetrapods.

## Results

### Phylogenetic analysis indicates that type I and type II hair keratin gene families have originated in stem tetrapods

To determine when in evolution genes of the type I and type II hair keratin gene families have appeared, we compared the organization of keratin gene loci (Fig. 1a, b) and performed phylogenetic analyses (Fig. 1c, d) of keratins of species representing the three main clades of amphibians, i.e. frogs (western clawed frog, *X. tropicalis*), salamanders (axolotl, *Ambystoma mexicanum*) and caecilians, together with keratins of the lungfish (*Protopterus annectens*) as the closest piscine relative of tetrapods[24]. We found that orthologs of human type I and type II hair keratins, denoted *krt34* and *krt59* in amphibians, exist in the clawed frog and the axolotl but not in the lungfish and in caecilians, which are limbless amphibians (Fig. 1). Orthology of *krt34* and *krt59 X. tropicalis* and axolotl with human hair keratins was supported by bootstrap values > 0.9 in molecular phylogenetics (Fig. 1c, d) and shared synteny of gene loci (Fig. 1a, b). The species distribution of hair keratin homologs suggests that the hair keratin subfamilies of type I

and type II keratins have originated in stem tetrapods and that they have been lost in caecilians (Fig. 1e).

### Homologs of hair keratins are components of claws in *Xenopus* frogs

For further studies of amphibian hair keratin expression, we used the western clawed frog (*X. tropicalis*) as a model. *X. tropicalis* is a diploid relative of the allotetraploid African clawed frog (*X. laevis*), in which the morphology and development of claws were studied previously[11]. Cornified claws develop on toes I, II, III of the hindlimbs, here collectively referred to as hindlimb inner (HI) toes. Toes IV, V of the hindlimbs, here referred to as hindlimb outer (HO) toes, and the toes of the forelimbs lack claws (Fig. 2a) in *X. tropicalis*. Claws ensheath the living epithelium from which they differentiate (Fig. 2b).

RT-PCR analysis showed that the hair keratin orthologs, *krt34* and *krt59*, are expressed in the clawed HI toes but neither in clawless HO toes nor in back skin of *X. tropicalis* (Fig. 2c, d). The corresponding proteins were highly abundant in cornified claws, as revealed by mass spectrometry (MS)-based proteomic analysis (Fig. 2f, g). By contrast, *X. tropicalis krt53,* which is associated with adult frog skin[8] and not related to hair keratins (Fig. 1c), was not enriched in claws (Fig. 2e, h).

### Co-expression in frog toes and promoter assays in transfected cells suggest regulation of amphibian hair keratin homologs by Hoxc13

Given the conserved expression of hair keratin homologs at the toe tips of frogs and mammals and loss of hair keratin homologs in limbless caecilians (Fig. 1a, b) and snakes[26], we hypothesized that hair keratin homologs of both amphibians and amniotes are regulated by Hoxc13, a transcription factor evolutionarily associated with the development of limbs[24]. As the contribution of Hoxc13 in the regulation of human hair keratins was already demonstrated previously[14], we focused on *X. tropicalis* as model for amphibians. mRNA in situ hybridization showed that *hoxc13* (Fig. 3a, d; Supplementary Fig. 1a, c), *krt34* (Fig. 3b, e; Supplementary Fig. 1b, d) and *krt59* (Fig. 3c, f) co-localize in the claw-forming epithelium of *X. tropicalis*. Next, we screened the promoter sequences of hair keratin homologs of clawed frogs for sequences similar to the binding sites of Hoxc13 that were previously identified and validated within human hair keratin promoters[14]. Several sites matching the Hoxc13 binding consensus sequence, TT(T/A)ATx(A/G)(A/G), were identified in the proximal promoters of *krt34* and *krt59* genes of *X. tropicalis* and *X. laevis* (Fig. 3g). These potential binding sites of Hoxc13 were located at similar distance from the TATA box as the validated binding sites in human hair keratin genes (Fig. 3g).

To test whether Hoxc13 is able to activate transcription from the proximal promoter regions of *krt34* and *krt59*, we co-transfected HEK293T cells with a Hoxc13 expression construct and a reporter vector harboring a luciferase reporter gene downstream of the proximal promoter sequences of *krt34* and *krt59* in which predicted Hoxc13 binding sites were either preserved as in wildtype (WT) genes or mutated. Hoxc13 activated transcription of both *krt34* and *krt59* promoter constructs (Fig. 3h, i). Mutations of putative Hoxc13 binding sites in *krt34* and *krt59* promoters significantly reduced luciferase reporter expression (Fig. 3h, i). Residual activation by Hoxc13 suggests that Hoxc13 binds at additional promoter sites besides those mutated in our experiment. Together, these data show that homologs of hair keratins, which are essential for hair and nail development in mammals[15,17], are expressed in claws of *X. tropicalis* and their promoters contain conserved and functional Hoxc13 binding sequences.

### Deletion of Hoxc13 abolishes the expression of hair keratin homologs and the formation of cornified claws in *X. tropicalis*

To test the role of Hoxc13 in the regulation of keratins in *X. tropicalis*, we generated Hoxc13-deficient frogs and determined their phenotype

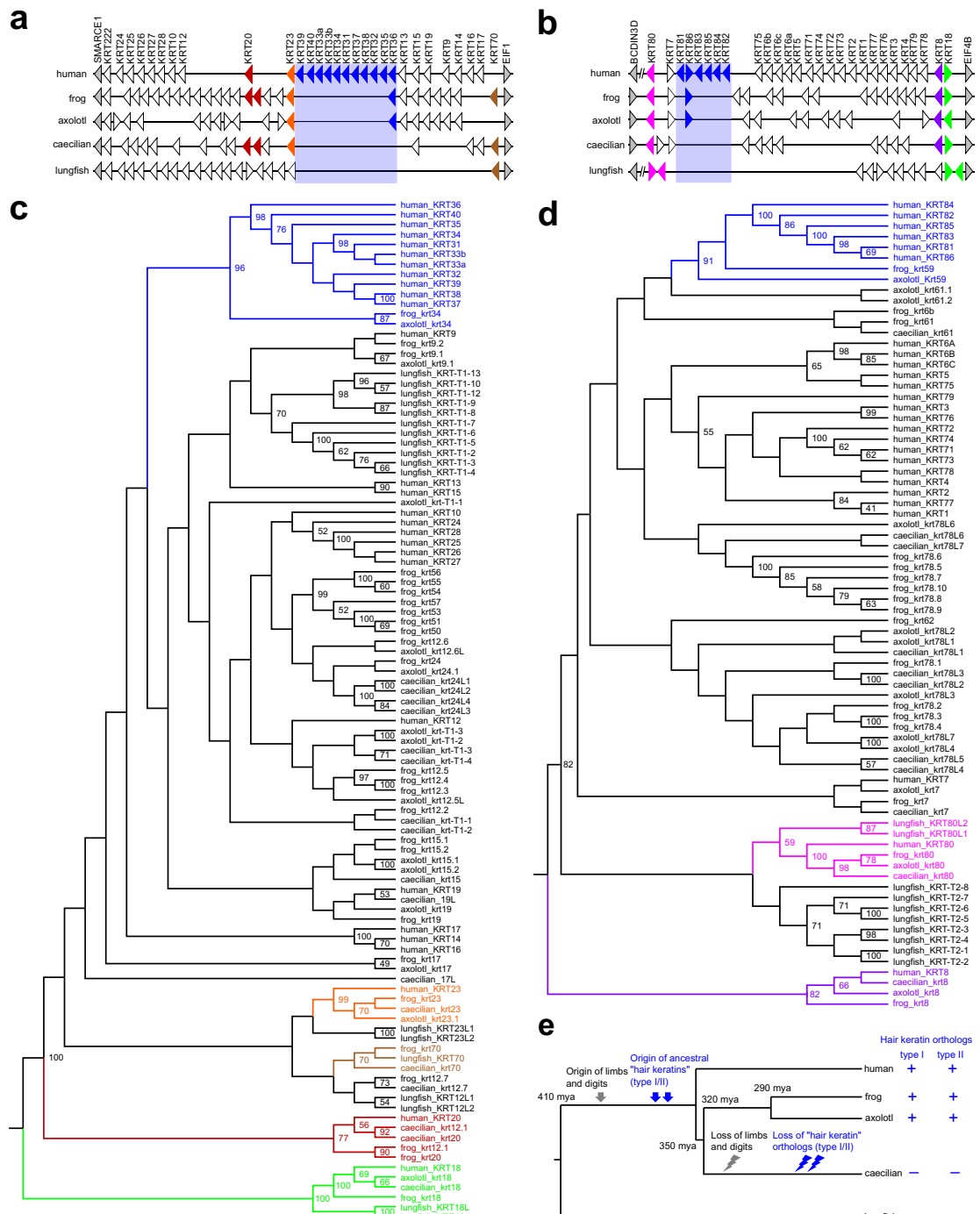

**Fig. 1 | Synteny analysis and molecular phylogenetics identify hair keratin homologs in amphibians. a**, **b** Schematic representation of the keratin type I (**a**) and type II (**b**) gene clusters of selected vertebrates: human, *Homo sapiens*; frog, *Xenopus tropicalis*; axolotl, *Ambystoma mexicanum*; caecilian, *Rhinatrema bivittatum*; lungfish, *Protopterus annectens*. Genes are shown as triangles pointing in the direction of transcription. Homologs of hair keratin genes are highlighted by blue shading. Note that *KRT18* is the only type I keratin gene located in the type II cluster. Slanted double lines indicate sites where genes are omitted for clarity. **c**, **d** Phylogenetic analysis of type I (**c**) and type II (**d**) keratins. Keratin orthologs in

amphibians and human or lungfish with bootstrap values > 50 (indicated in **c**, **d**) are shown with matching colors in panels (**a**) and (**c**) (type I) and (**b**) and (**d**) (type II). Nodes without numbers have bootstrap values < 50. Note the strong support (bootstrap >90) for orthology of amphibian *krt34* (**c**) and *krt59* (**d**) with human hair keratins (highlighted in blue) of the respective keratin type. Only fully sequenced keratin genes were included in this analysis. **e** Model for the evolution of hair keratin homologs in relation to the evolution of amphibians[54]. mya, million years ago; +, present; -, absent.

in comparison to control frogs. A double-strand break was introduced by CRISPR-Cas9-mediated genome editing[27] at the codon for amino acid residue 80 of Hoxc13 to cause nucleotide deletions or insertions of various lengths upstream of the homeobox domain (amino acid residues 239-292), which is essential for DNA binding and activation of

transcription (Supplementary Fig. 2). Out-of-frame (frame-shift) mutations were predicted to abrogate protein function. By contrast, short in-frame deletions and substitutions of single amino acid residues at this site were predicted to be compatible with the function of Hoxc13 as a transcription factor, thereby providing negative controls

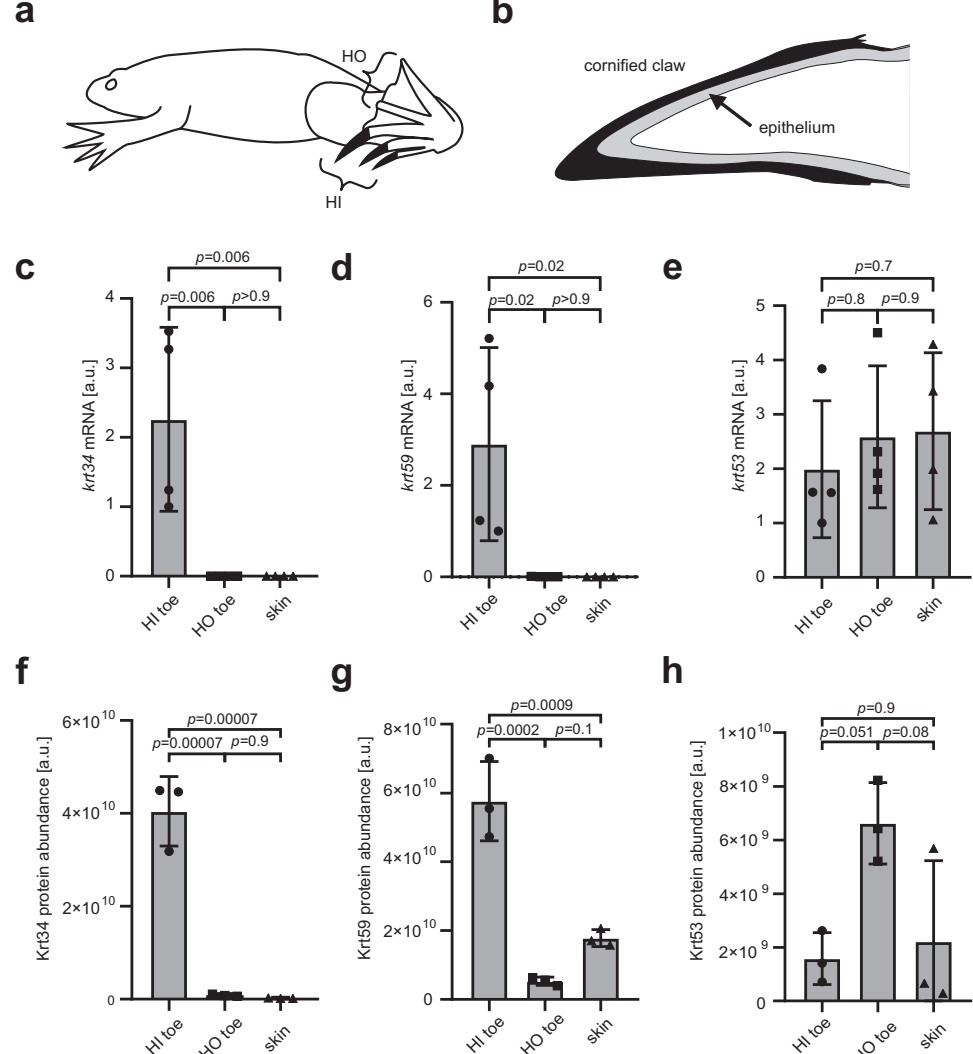

**Fig. 2 | Expression of hair keratin homologs of *Xenopus tropicalis* is associated with cornified claws. a** Schematic depiction of a clawed frog (*X. tropicalis*) bearing cornified claws on the hindlimb inner (HI) toes (toes I, II, III) and no claws on the hindlimb outer (HO) toes (toes IV, V) and on toes of the forelimbs. **b** Schematic of a frog clawed toe tip. The term "claw" refers to the cornified claw that is formed by the differentiation of the epithelium on the tip of the toe. **c–e** Quantitative RT-PCR analysis of mRNA expression of *krt34* (**c**), *krt59* (**d**), *krt53* (**e**) relative to the house-keeping gene, *eef1a1*. Mass-spectrometric quantification of Krt34 (**f**), Krt59 (**g**) and Krt53 (**h**). a.u., arbitrary units. RNA and protein were sampled at an age of 7 months. Statistics was calculated by one-way ANOVA for $n = 4$ (**c–e**) and three (**f–h**) biological replicates, respectively, in each of 3 groups. Bars and error bars indicate means and standard deviations, respectively.

for the following investigations of the phenotypes. Two-cell stage embryos of *X. tropicalis* were targeted by CRISPR-Cas9-induced mutagenesis to generate mosaic frogs carrying different mutations in different parts of the body including germ cells. By crossing mosaic mutant frogs, we obtained frogs that carried one or two mutant alleles of *hoxc13* in all cells of the body (Fig. 4a).

Frogs with out-of-frame deletions in both *hoxc13* alleles, here referred to as knockout (KO) frogs, developed normally except for one striking phenotype, the absence of claws (Fig. 4b–e). By contrast, frogs with one out-of-frame mutant allele (heterozygotes) and frogs with in-frame deletions (3 or 6 nucleotides) in both *hoxc13* alleles displayed normal development of macroscopically visible claws on HI toes (Fig. 4a). Furthermore, some of the mosaic mutant frogs lacked claws on one or both hind-limbs. Sequencing of *hoxc13* showed two out-of-frame alleles in clawless HI toes and the presence of one allele encoding a functional Hoxc13 protein in clawed HI toes of a mosaic mutant (Supplementary Fig. 3). Histological analysis confirmed the presence of darkly pigmented cornified claws on HI toes of WT frogs (Fig. 4f) and their absence on the HI toes of KO frogs (Fig. 4g). These

phenotyping data demonstrated that biallelic out-of-frame mutations of *hoxc13* abolished, either directly or indirectly, the differentiation of toe tip keratinocytes into cornified claws whereas single alleles of wildtype or in-frame-mutant *hoxc13* sufficed to induce claw formation. As off-target effects of CRISPR-Cas9 genome editing are equally likely in frogs with in-frame mutations and frogs with out-of-frame muta-tions of *hoxc13*, the claw-loss phenotype can be clearly attributed to the direct inactivation of *hoxc13*.

The impact of *hoxc13* mutations on the expression of hair keratin homologs was determined by quantitative RT-PCR analysis (Fig. 4h, i). The inactivation of Hoxc13 abrogated the expression of both *krt34* (Fig. 4h) and *krt59* (Fig. 4i) in HI toes. RNA-seq analysis showed that *krt34* and *krt59* were more strongly decreased than any other keratin in Hoxc13-deficient (clawless) HI toes as compared to HI toes in which Hoxc13 (either wildtype or carrying an R80M substitution) was expressed (Supplementary Fig. 4). By contrast, *krt78*.3 and *krt78*.4 were upregulated in the absence of Hoxc13, suggesting that an alternative keratinization program is activated in clawless HI toes (Supplementary Fig. 4). Collectively, these data demonstrate that Hoxc13 is essential for

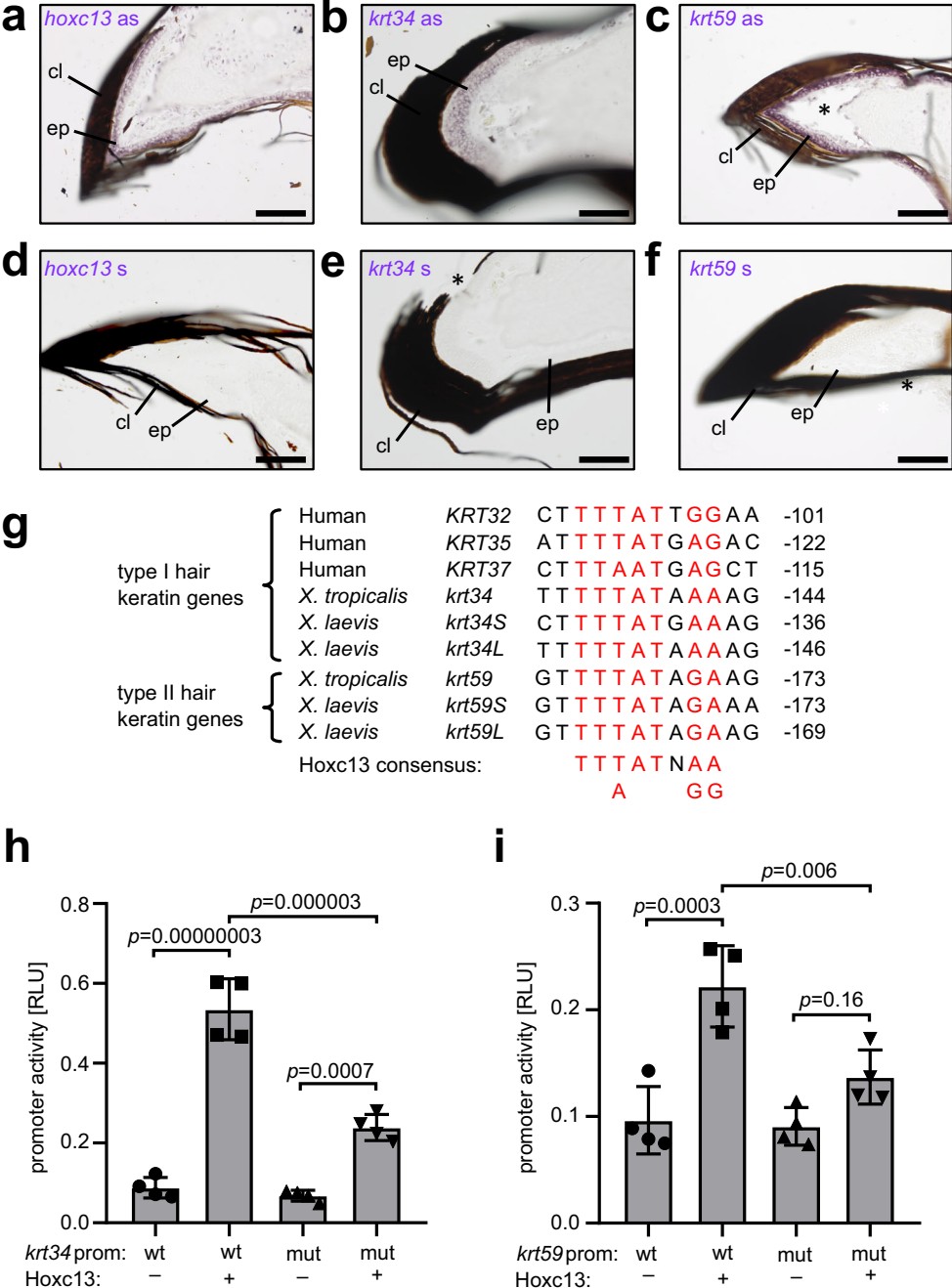

**Fig. 3 | Co-expression with *krt34* in clawed toes and promoter assays suggest Hoxc13 as regulator of hair keratin expression. a–f** mRNA in situ hybridization using *hoxc13* (**a**), *krt34* (**b**), and *krt59* (**c**) antisense (as) probes on sections of HI toes of *X. tropicalis* sampled at an age of 4 months (3 replicates each). Hybridizations with sense (s) probes (**d–f**) served as negative controls. Artefacts such as the detachment of the cornified claw during tissue sectioning (*) are indicated. cl, claw; ep, epidermis. Scale bars: 100 μm. **g** Consensus Hoxc13 binding sites in the proximal promoters of human and frog hair keratin homologs. The distance to the TATA box

(number of nucleotides) is indicated on the right. The consensus sequence of Hoxc13 binding sites in human hair keratin promoters[19] is shown below the sequence alignment. **h, i** Luciferase activity of cells co-transfected with a Hoxc13 expression vector and a luciferase reporter under the control of the *krt34* (**h**) or *krt59* (**i**) promoter (prom) containing wildtype (wt) or mutated (mut) Hoxc13 binding sites. Statistics was calculated by one-way ANOVA for $n = 4$ biological replicates in each of 4 groups. Bars and error bars indicate means and standard deviations, respectively. RLU relative light units.

the expression of hair keratin homologs and the development of claws in western clawed frog, *X. tropicalis*.

**The expression of *hoxc13* and hair keratin homologs is conserved in the axolotl**

As orthologs of *krt34* and *krt59* are also present in salamanders (Fig. 1), we investigated their expression pattern in the axolotl. Although the axolotl lacks claws, its toe tips are covered by a brownish cornified cell

layer (Fig. 5a–d). RT-PCR analysis of keratins showed that *hoxc13* and the hair keratin orthologs, *krt34* and *krt59*, are expressed in the toes but not in the skin of the belly, whereas other epithelial keratins are expressed independently of Hoxc13 at both body sites (Fig. 5e). mRNA in situ hybridization demonstrated expression of *krt34* in the epithelium of the toe tips of the axolotl (Fig. 5f). These data show that the expression of *hoxc13* and homologs of hair keratins is conserved in the toe tips of representative species of anurans and salamanders.

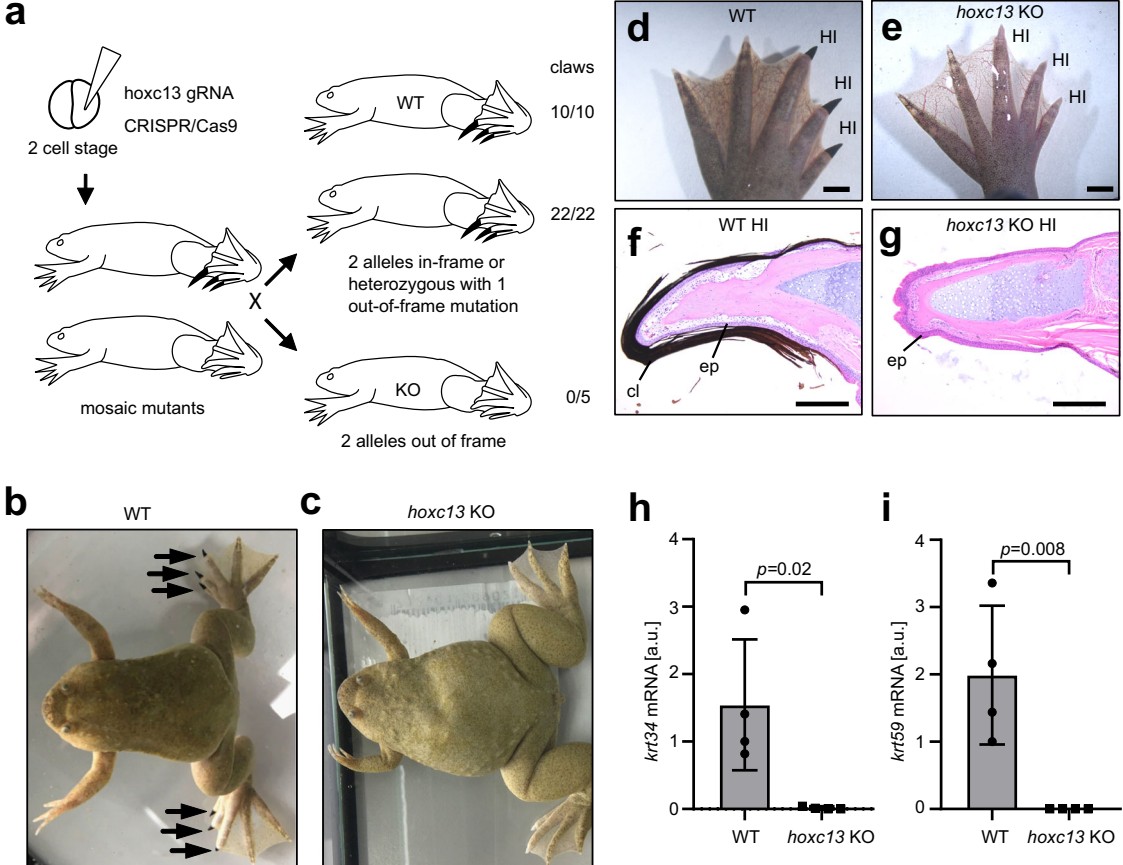

**Fig. 4 | *hoxc13* knockout frogs lack claws and do not express hair keratins homologs.** **a** Schematic depiction of *hoxc13* knockout (KO) in *X. tropicalis* frogs. The number of animals bearing claws is shown on the right. Macroscopic appearance of a wildtype (WT) (**b**) and a KO (**c**) frog. Arrows point to claws. **d**–**g** Hindlimb phenotypes (**d**, **e**) and histology (**f**, **g**) of hindlimb inner (HI) toes of WT (**d**, **f**) and KO (**e**, **g**) frogs. Images in b-g are representative for at least 3 biological replicates. cl claw, ep epidermis. Scale bars: 2 mm (**d**, **e**), 200 μm (**f**, **g**). Quantitative RT-PCR analysis of *krt34* (**h**) and *krt59* (**i**) in HI toes of WT and KO. Statistics was calculated by unpaired two-tailed *t*-test. *n* = 4 biological replicates for WT and KO. Age of the frogs: 7 months. a.u., arbitrary units. Bars and error bars indicate means and standard deviations, respectively.

Together with the confirmation that *hoxc13* and homologs of hair keratins are also expressed in the clawed toes of green anole lizards (Supplementary Fig. 5) and the well-established co-expression of *hoxc13* and hair keratins in nails of mammals, these data lead us to conclude that the expression of *hoxc13* and homologs of hair keratins in toe tips was inherited from a common ancestor of tetrapods (Fig. 6).

## Discussion

This study reveals an unexpected conservation of the molecular composition and transcriptional regulation of cornified skin appendages in frogs and humans, and suggests a scenario for the evolution of hair. In contrast to the previously held assumption that claws of clawed frogs and claws of amniotes have evolved independently[11], the present study indicates that they are homologous. Claws of *Xenopus* form by the accumulation of layers of cornified keratinocytes whereas keratinocytes in the claws of amniotes lack this organization[11,12]. Unexpectedly, we found a single layer of cornified pigmented keratinocytes also on the toe tips of the axolotl, suggesting the existence of a similar mode of keratinocyte differentiation, but without thickening of the cornified layer. Conservation of Hoxc13 and keratin expression indicate that the cornified epithelium of axolotl toes is homologous to claws. Our data show that Hoxc13 is essential for the expression of homologs of both type I and type II hair keratins in *Xenopus* claws, which is equivalent to the critical role of Hoxc13 in controlling the expression of hair keratins in human nails and hair[15]. The conservation of critical genes indicates that claws of amphibians, claws of amniotes

and hair of mammals have evolved from a common ancestral skin structure.

We propose the following scenario for the evolution of claws and hair (Fig. 6). Hoxc13 was present in the last common ancestor of all jawed vertebrates with a primordial function in regulating the development of the tail fin. In the last common ancestor of lungfish and tetrapods, expression of Hoxc13 appeared, together with other distal genes of Hox gene clusters[28], in paired fins[24]. The latter evolved into limbs of tetrapods and, accordingly, Hoxc13 was expressed in the distal phalanges of primitive tetrapods. At this stage of evolution, the promoters of a subset of keratin genes acquired Hoxc13 binding sites, thereby facilitating the specific expression of these keratins on toes. This expression pattern of specialized keratins allowed the adaptive evolution of epithelial structures for toe-associated functions, such as the protection against mechanical stress, the support of locomotion as provided also by adhesive toe pads of modern tree frogs, or the capture of prey as facilitated by claws of several species of amniotes. The original function of the primordial "hair keratins" as "toe keratins" is conserved in toe pads of tree frogs[10], toe tips of the axolotl (this study), claws of clawed frogs (this study), claws of lizards[6] and nails of humans[15]. As part of a complex evolutionary transition that is not yet understood, the evolution of *Hoxc13* expression at other sites of the integument, besides toes, redirected the expression of the Hoxc13-dependent toe keratins and contributed to the evolutionary innovation of hair shafts in which the primordial toe keratins became the main structural components, i.e. hair keratins.

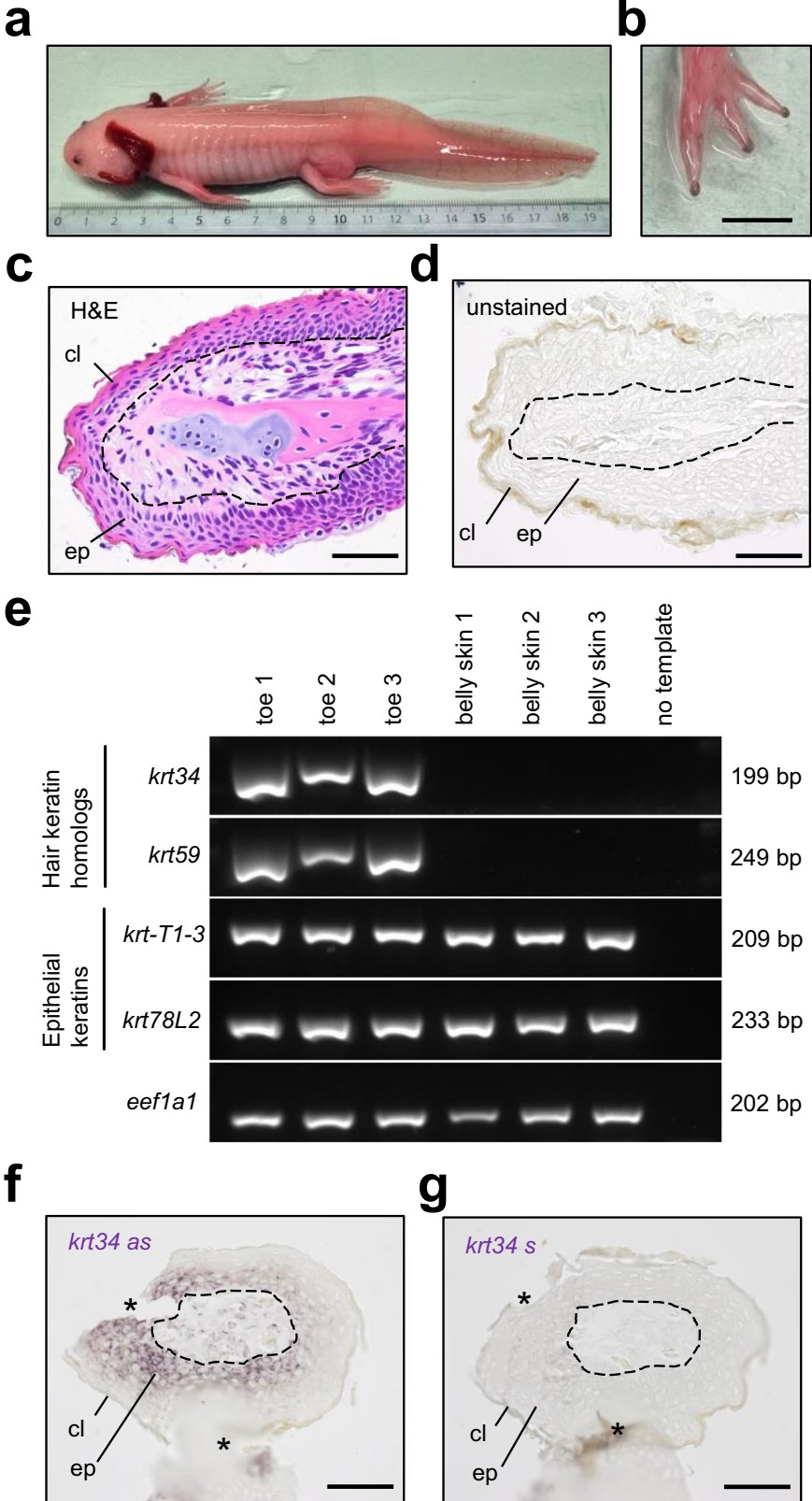

**Fig. 5 | Hoxc13 and hair keratin homologs are expressed at the toe tips of the axolotl. a** Macroscopic view of an albino axolotl. **b** View of the pigmented toe tips. Sections of toe tips were stained with hematoxylin and eosin (**c**) or were left unstained (**d**). **e** RT-PCR analysis of *hoxc13*, hair keratin homologs (*krt34* and *krt59*), epithelial keratins (*krt-T1-3* and *krt78L2*) and the housekeeping gene *eef1a1* in toes and belly skin. **f** In situ hybridization of *krt34* mRNA using an anti-sense (as) probe.

**g** Negative control experiment using a *krt34* sense (s) probe. Images are representative for *n* = 3 adult individuals of albino axolotl. Artefacts such as the detachment of tissue during tissue sectioning (*) are indicated. cl cornified layer, ep epidermis. Dashed lines indicate the junction of epidermis and dermis. Scale bars: 5 mm (**b**), 100 μm (**c**, **d**, **f**, **g**).

This scenario is simplified by focusing on the evolution of hair keratins and their regulation by Hoxc13. A more comprehensive model for the evolution cornified skin appendages will need to address other important aspects, such as epithelial-mesenchymal signaling and morphogenesis[1,2,29]. For example, sonic hedgehog (Shh) is implicated in the evolution of skin appendages and plays a critical role in the morphogenesis of hair follicles[30–32]. Interestingly, the inhibition of Shh signaling in late stages of limb development allows the formation of claws on digits IV and V (HO toes) of *Xenopus tropicalis*[33]. The effects of Shh signaling on the expression of hair and claw keratins are likely indirect, whereas experimental data obtained in cultured cells indicate that Hoxc13 directly regulates the expression of human hair keratins[14] and *Xenopus* claw keratins (Fig. 3h, i). However, additional indirect effects of Hoxc13 on keratin genes are possible in vivo. While the regulation of hair keratin homologs by Hoxc13 appears to have evolved in stem tetrapods, the high content of cysteine residues in hair keratins[6,34] is not shared by their orthologs in amphibians[9], suggesting that the initial evolution of keratin gene regulation preceded the evolution of cysteine-dependent crosslinking of hair keratin homologs. The scenario for the evolution of claws and hair (Fig. 6) does not imply that Hoxc13 and hair keratins were involved in the evolution of all types of cornified skin appendages. In particular, the molecular architecture of feathers, which involves sauropsid-specific cysteine-rich keratins, beta-keratins (also known as corneous beta proteins) and other epidermal differentiation proteins[9,35], has likely evolved independently from claws and hair[9,36], and the role of Hoxc13 in the control of feather keratins is not known.

The evolutionary model depicted in Fig. 6 suggests further directions of research. For example, the mechanism of the shift of Hoxc13 expression from toes to newly evolving hair follicles at other body sites is currently unknown. This evolutionary transition has probably involved intermediate stages, such as hair fibers as organs of tactile sensation[37] or filiform papillae of the tongue[38]. In line with these hypotheses, Hoxc13 is expressed in both vibrissae and lingual filiform papillae of the mouse[17]. At the genetic level, mammal-specific enhancer elements of the Hoxc13 gene have contributed to evolutionarily important changes of expression[25]. The roles of other protein components besides hair keratins and the roles of other regulatory factors besides Hoxc13 remain to be integrated into the scenario for the evolution of hair and nails.

The evolution of Hoxc13 from a regulator of toes to a regulator of toe keratins links the evolution of body appendages to the evolution of skin appendages in tetrapods. Furthermore, the primordial function of Hoxc13 in embryonic development evolved into a new function in cell differentiation within an adult tissue. Skin appendages, such as claws, nails and hair are regenerating throughout life to compensate for abrasion and damage caused by the direct contact to the environment. Thus, the cooption of a regulator of development has contributed to the evolution of tissue regeneration in the adult organism.

## Methods
### Animals
*Xenopus tropicalis* animals were kept at 25–27 °C in an aquatic system on a biofilter. All experiments on *X. tropicalis* were executed in accordance with the guidelines and regulations of Ghent University, Faculty of Sciences, Ghent, Belgium. Approval of the project was obtained by the Ethical Committee for Animal Experimentation, Ghent University, Faculty of Sciences (approval number EC2023-041). Additionally, *X. tropicalis* frogs were purchased from Interaquaristik.de Shop (Biedenkopf-Breidenstein, Germany). These animals were killed and tissues were sampled in accordance with the guidelines of the Ethics Committee of the Medical University of Vienna. Both male and female animals were investigated. Axolotls were generously provided by Elly Tanaka, Research Institute of Molecular Pathology (IMP),

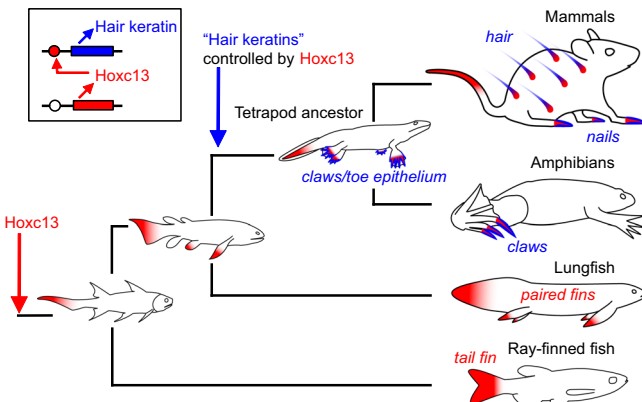

**Fig. 6 | Model for the evolutionary origin of Hoxc13 and hair keratin-dependent cornified skin appendages.** The evolutionary changes in the expression pattern of Hoxc13 and the appearance of the regulatory link between Hoxc13 and "hair keratins" were inferred from data obtained in extant vertebrates and their known phylogeny. "Hair keratins" refers to human keratins KRT31-KRT40 and KRT81-KRT86 and their orthologs in other species. Red and blue shading mark sites of expression of Hoxc13 and hair keratin homologs, respectively.

Vienna, Austria. cDNA from two adult individuals of *Anolis carolinensis* was prepared in a previous study[6].

### Generation of *hoxc13*⁻/⁻ frogs
A sgRNA targeting a site in the *X. tropicalis* hoxc13 coding sequence upstream of the homeodomain (GenBank accession number NC_030678.2, nucleotides 146200810-146200832) was designed using CRISPRScan (https://www.crisprscan.org/) and Indelphi (https://indelphi.giffordlab.mit.edu/) algorithms. The sgRNA was synthetized as previously described[27].

Natural matings were set up as described previously[27]. Embryos were collected and de-jellied in 2% L-cysteine (catalog number: CALB243005-1, Calbiochem) in 0.1x Marc's Modified Ringer Solution (MMR) adjusted to pH 8.0 (10x MMR stock: 1 M NaCl, 20 mM KCl, 20 mM CaCl$_2$, 10 mM MgCl$_2$, 50 mM HEPES, pH 7.8), washed and submerged in 0.1X MMR/6% Ficoll (catalog number: F8016, Sigma-Aldrich). Two-cell stage embryos were bilaterally injected with 1 nl of a mix of sgRNA and recombinant NLS-Cas9-NLS (VIB Protein Service Facility, UGent) to generate mutant mosaic F0 offspring. The editing efficiency was determined by deep amplicon sequencing of DNA extracted from embryos after injection as described previously[39]. The F1 generation was obtained via subsequent breeding of the mosaic F0 founders. Genotyping was done by either high-resolution melting analysis (HRMA) analysis as previously described[40] and/or Sanger sequencing by Mix2Seq (Eurofins Genomics). All genotyping was validated by targeted amplicon sequencing and BATCH-GE analysis[41] or CRISPResso2 analysis (version: 2.2.1)[42]. Primers for HRMA analysis were HRMAF1hoxc13v1, 5'-CTAGTGCCATTGGGTCTCAC-3'; HRMAR1-hoxc13v1, 5'-AGAGCCTGCAGCCGTAATAG-3'; HRMAF2hoxc13v1, 5'-CACGGACATTACCTCTCCAG-3'; HRMAR2hoxc13v1, 5'-AGTTGACGTTGTGGGACTGA-3'; HRMAF1hoxc13v2, 5'-AGTAGTGCCAGGCATCAGC-3'; HRMAR1hoxc13v2, 5'-GGATTGGTCCTTTGAGCAGT-3'. To determine the genotype of toes in F0 mosaic mutant frogs, hoxc13 was amplified from cDNA preparations using the primers HRMAF1hoxc13v1 and HRMAR1hoxc13v1. The products were ligated into a pGEM-T Easy vector (catalog number: A1380, Promega) which was amplified in *E. coli* (catalog number: L2001, Promega). Using the same primers, hoxc13 was also amplified from toe cDNAs of wildtype frogs and frogs carrying two hoxc13 KO alleles. PCR products and inserts of plasmids were subjected to Sanger sequencing (Microsynth). Frogs containing two alleles with out-of-frame mutations, leading to premature termination

of the coding sequence upstream of the homeodomain-encoding region were designated as *hoxc13* knockout frogs. Wildtype frogs, heterozygous frogs containing one wildtype and one out-of-frame allele of *hoxc13*, and frogs containing two alleles with in-frame deletions were investigated as controls.

## Tissue sampling and fixation
Adult *X. tropicalis* frogs were euthanized with 0.15% benzocaine (catalog number: 94-09-7, Sigma-Aldrich). Hindlimbs were examined before tissue sampling and images were taken with a Zeiss Lumar V12 stereoscope and a Zeiss AxioCam MRc. Tissue for RNA extraction was dissected and placed immediately in RNAlater (catalog number: AM7020, Invitrogen). Tissue for histological analysis was incubated in 7.5% neutral-buffered formalin (NBF) (catalog number: 18727.00500, Morphisto) overnight. Formalin-fixed toes were decalcified by incubation in a solution containing 5% formic acid (catalog number: 695076, Honeywell-Fluka) and 0.2 M EDTA, adjusted to pH 8.0 with sodium hydroxide, for three days. Afterwards samples were washed intensively with water and incubated in 5% sodium sulfate to neutralize the samples. Samples were further washed for one day in water and subsequently treated with 10% thioglycolate in 7.5% NBF for two days, followed by a last washing step with water and dehydration. The samples were embedded in paraffin and stored for further analysis.

## Histology
Sections of formalin-fixed tissues were prepared with a Microm HM 335E microtome at a thickness of 5 μm and collected in a water bath set to 42 °C. The sections were subjected to hematoxylin and eosin (H&E) staining and photographed with an Olympus BX63 light microscope equipped with a UC-90 camera and the software cellSense Dimension (version: 2.3.18987.0).

## Preparation of riboprobes
Riboprobe templates that anneal at the end of the coding region and the 3′-untranslated regions of *X. tropicalis hoxc13* and *krt34* were amplified by RT-PCR from HI toe cDNA. The PCR products were ligated into a pGEM-T Easy vector (catalog number: A1380, Promega) and plasmids containing the correct insert were amplified in *E. coli* (catalog number: L2001, Promega). T7 RNA polymerase was used for in vitro transcription of the riboprobes. Detailed information is provided in the Supplementary Methods.

## In situ hybridization
mRNA in situ hybridization was performed on tissue sections of formalin-fixed paraffin-embedded HI toes. The samples were incubated with antisense probes that targeted either *hoxc13* or *krt34*. The corresponding sense probes were used as negative controls. Detailed information is provided in the Supplementary Methods.

## RNA isolation and quantitative RT-PCR
RNA was isolated from tissues using Trizol (catalog number: 30-2010, VWR) and a Precellys homogenizer (VWR) following the instructions of the manufacturer. The RNA was reverse-transcribed with the qScript® cDNA synthesis kit (catalog number: 95047-025, Quantabio). cDNAs from *X. tropicalis* tissues were subjected to real-time PCR with primers (Microsynth) specific for *krt34* (forward: 5′-GTTACGGCGGTCA ACTTTCC-3′ and reverse: 5′-CAAGGCAATCAGGCGAAGTT-3′), *krt59* (forward: 5′-CTGCTAGAAGGGGAGGAAGG-3′ and reverse: 5′-TGTT TCATCCTCAGGCTCCA-3′), *krt53* (forward: 5′-GCTGGATAAACT AGGGGCCA-3′ and reverse: 5′-AGAAGTCAACGATCCCTGGA-3′), and *eef1a1* (forward: 5′-CTGAAGTCTGGTGATGCTGC-3′ and reverse: 5′-GG AGCTGGCAAAGTCACAAA-3′). Amplifications were performed on a Light Cycler® 480 II (Roche) and involved initial denaturation at 95 °C for 10 min and 50 cycles of 95 °C for 10 s, 60 °C for 10 s and 72 °C for 15 seconds with a ramp rate of 2.2 °C per second. Quantities of target

mRNAs relative to transcripts of the housekeeping gene, *eef1a1*[43], were calculated according to a published model[44]. Axolotl cDNAs were subjected to PCRs with primer pairs that anneal to *hoxc13* (forward: 5′-GGACGGACAGGTCTACTGC-3′ and reverse: 5′-GGAGCTGGAG AAGGAGTACG-3′), *krt34* (forward: 5′-AACCAGATCCAGGGCTTGAT-3′ and reverse: 5′-TTCCGGAGGCAAAGATGTCT-3′), *krt59* (forward: 5′-AA GCTGGGTCTGGACATTGA-3′ and reverse: 5′-GCGGAAGCAGATT TAGCACA-3′), *krt-T1-3* (forward: 5′-AAGATCCGAGACTGGTACCA-3′ and reverse: 5′-GAGAATGAGCTGGCCATGTG-3′), *krt78L2* (forward: 5′-G ATTGCCGACACCTCTGTTG-3′ and reverse: 5′-ACTGAACCGCATGA TCCAGA-3′), *eef1a1* (forward: 5′-TGCTCACATCGCTTGCAAAT-3′ and reverse: 5′-TGCGTGACATGAGACAGACT-3′). cDNAs of the green anole lizard (*Anolis carolinensis*)[6] were subjected to PCR with primers specific for *hoxc13* (forward: 5′-CTTTGCCAGCTCCTACCAAG-3′ and reverse: 5′-AAGTGAGCAGCTACCGTAGG-3′), *krt36L/HA1* (forward: 5′-A ATAAGGTCCTGGATGAGATG-3′ and reverse: 5′-CTGATCAGCAATGT AGAGGCC-3′), *krt84L/HB1* (forward: 5′-CTCTCAAAGATGCCAAGTGC-3′ and reverse: 5′-TGTGATCCCGATCCTTGCATT-3′)[6], and *eef1a1* (forward: 5′-TTGCCACACTGCCCATATTG-3′ and reverse: 5′-AGGCAGTTG ACAAGAAAGCG-3′).

## Analysis of proteins
Tissue samples for proteomic analysis were dissected freshly and placed immediately in 200 μl lysis buffer, which consisted of 30 mM Tris, 7 M urea (catalog number: 0568, VWR), 2 M thiourea (catalog number: T7875, Sigma-Aldrich), 4% CHAPSO (catalog number: 28304, Pierce) and 0.2 M dithiothreitol (DTT). After incubation at 70 °C for 3 h the samples were homogenized with a homogenizer (Precellys, VWR) and centrifuged at 18,000 g for 15 min at 4 °C. The supernatant was collected and processed for mass spectrometry-based proteomics as described in the Supplementary Methods.

## Hoxc13 binding site prediction
Potential Hoxc13 binding sites in the *krt34* and *krt59* promoter regions were identified using the Scan tool at the JASPAR website (https://jaspar.genereg.net/, last visited on 22.05.2023) with a relative profile score threshold of 70%. The search for Hoxc13 binding sites was narrowed down by searching specifically for the sequences matching the consensus Hoxc13 binding motif that was defined and validated in human hair keratin promoters[14]. Two sites of the *krt34* promoter (see the Methods section "Construction of vectors for the promoter activation assay") and one site of the *krt59* promoter (see the Methods section "Construction of vectors for the promoter activation assay") were selected for testing in a promoter activation assay (see below). Note that some of the potential Hoxc13 binding sites identified by JASPAR were not altered in the mutated reporter plasmids.

## Construction of vectors for promoter activation assays
The complete coding sequence of *X. tropicalis hoxc13* (GenBank accession number XM_002936645.5, nucleotides 78-998), flanked by the sequence TGTAAAACGACGGCCAGTGGATCCGCCACC on the 5′-side and TCTAGAGGTCATAGCTGTTTCCTG on the 3′-side, was synthesized as G-block DNA fragment (Integrated DNA Technologies, IDT). This fragment was amplified using M13 primers (forward: 5′-TGT AAAACGACGGCCAGT-3′ and reverse: 5′-CAGGAAACAGCTATGACC-3′) and cloned at *BamHI* and *XbaI* restriction sites in the pCS2+ plasmid[45]. The correct sequence of the construct was verified by whole plasmid sequencing (Plasmidsaurus).

For the generation of reporter plasmids, G-blocks (IDT) containing wild-type and mutated proximal promoter regions of *X. tropicalis krt34* and *krt59* (Supplementary Fig. 6) were inserted into pGL4.10[luc2], linearized by digestion with *XhoI* and *EcoRV*, using the Genbuilder® cloning kit (Genscript) according to manufacturer's instructions. The sequences of the constructs were verified by Sanger sequencing after amplification with the primers 5′-gaatcgatag

tactaacatacgctctc-3′ and 5′-GATCTGGTTGCCGAAGATGGG-3′. All plasmids generated in this study were deposited in the BCCM/GeneCorner Plasmid Collection, Ghent University, Ghent, Belgium.

### Transient transfection and luciferase reporter-based promoter activation assays

HEK293T cells were cultured in Dulbecco's Modified Eagle Medium (DMEM, Gibco, Waltham, MA, USA), supplemented with 10% fetal calf serum. HEK293T cells were seeded at a density of $4 \times 10^4$ cells in 24-well plates and transfected with a plasmid DNA mix (1000 ng total DNA per well) using the calcium phosphate method. The plasmid DNA mix consisted of 200 ng pGL4.10[luc2]-Krt[x] (with x one of the keratin promoter constructs), 200 ng of either pCS2+XtHoxc13 or the empty pCS2+ vector, 100 ng PB-RL2(PYL9) and 500 ng pCS2+. PB-RL2(PYL9) encoding Renilla luciferase was co-transfected as an internal control. Six hours after transfection, the medium was replaced by fresh medium and the transfected cells were maintained in culture overnight.

The krt-promoter luciferase assays were performed using the Dual-Luciferase® Reporter Assay System (Promega) according to manufacturer's guidelines. The transfected HEK293T cells were lysed in passive lysis buffer 24 h post transfection, 50 μL LARII was added to 20 μL lysate and firefly luminescence signals were measured. Afterwards 50 μL Stop&Glo® mix was added, and Renilla luminescence signals were again measured for standardization. All luminescence signals were measured using the GloMax 96 microplate luminometer (Promega).

### Molecular phylogenetics

Keratin sequences were collected from sequence databases listed in Supplementary Tables 1 and 2. Multiple sequence alignments of the intermediate filament rod domains were created with aliview[46] (Source Data file). Prottest (version 3.0)[47,48] was used to calculate the amino acid substitution model. All available matrices and models with rate variation among sites were included. The Akaike information criterion was used to assess the likelihood of the predicted models[49]. The best suited amino acid substitution model for the phylogeny was LG[50]. Maximum likelihood tree and bootstrap analysis was done using PhyML (Version 20120412). Tree topology, branch length, and rate parameters were optimized according to a published protocol[51]. Phylogenetic trees were visualized with FigTree (version: 1.4.3; http://tree.bio.ed.ac.uk/software/figtree/, last accessed on 2 June 2023) and edited with Inkscape (version: 1.0.0.0; https://inkscape.org/de/, accessed on 2 June 2023).

### RNA-seq

RNA was extracted from HI toes of two wildtype frogs and two hoxc13-KO frogs and separately from HI toes of genetically distinct hindlimbs of a mosaic mutant frog. In one leg of the mosaic mutant Hoxc13 R80M, considered equivalent to WT, was expressed, whereas expression of Hoxc13 was abrogated by frame-shift mutations in both hoxc13 alleles in the other leg (Supplementary Fig. 3). The RNA was subjected to RNA-seq analysis according to a published protocol[52] with modifications. The detailed experimental procedure and the differential gene expression analysis is documented in the supplement.

### Statistics

For the comparison of gene expression levels in 3 wildtype tissues, samples from 4 X. tropicalis frogs were investigated (Fig. 2c–e). Protein abundance in tissues (Fig. 2f–h) was determined by MS-based proteomics in samples from 3 frogs for each tissue. Statistical significance of differences was tested by one-way ANOVA using GraphPad Prism 8 (version 8.0.1) (Fig. 2c–h). For the comparison of gene expression levels in HI toes of wildtype and hoxc13 knockout frogs, 4 samples per genotype were investigated (Fig. 4h, i), and the statistical significance of differences was tested by an unpaired two-tailed t-test using

GraphPad Prism 8 (version 8.0.1). The luciferase reporter assay (Fig. 3f, g) was repeated four times with three technical replicates each. The statistical significance of differences was determined by one-way ANOVA.

### Reporting summary

Further information on research design is available in the Nature Portfolio Reporting Summary linked to this article.

## Data availability

The mass spectrometry-based proteomic data generated in this study have been deposited into the ProteomeXchange Consortium via the PRIDE[53] partner repository (https://www.ebi.ac.uk/pride/) under accession code PXD041765. The RNA-seq data generated in this study have been deposited on NCBI under accession code PRJNA1055414. Accession codes/sources for publicly available original sequence data used in this study are provided in the Supplementary Information. All data needed to evaluate the conclusions in the paper are present in the paper and/or the Supplementary Materials. Unique biological materials that support the findings of this study are readily available from the corresponding authors upon request. The multiple sequence alignments for phylogeny generated in this study are provided in the Source Data file. The X. tropicalis reference genome UCB_Xtro_10.0 used as reference database for the RNA-seq analysis is available on NCBI under accession number GCA_000004195.4. Source data are provided with this paper.

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

## Acknowledgements

This research was supported by the Core Facility Genomics of the Medical University of Vienna, a member of Vienna Life-Science Instruments (VLSI), and by the VetCore Facility (Proteomics) of the University of Veterinary Medicine Vienna. We thank Florian Gruber, Medical University of Vienna, for helpful discussions and Prof. Elly Tanaka, Research Institute of Molecular Pathology (IMP), Vienna BioCenter, for providing axolotls. This research was funded in part by the Austrian Science Fund (FWF): grant-DOI: 10.55776/P32777, grant-DOI: 10.55776/P36596 received by L.E. For open access purposes, the authors have applied a CC BY public copyright license to any author accepted manuscript version arising from this submission. M.C. received a PhD fellowship from the Research Foundation—Flanders (FWO-Vlaanderen). Research was supported by the Concerted Research Actions from Ghent

University (BOF15/GOA/011 and BOF20/GOA/023) received by K.V. and E.D.B.

## Author contributions
M.C., A.P.S., E.D.B., W.D., E.T., K.V. and L.E. designed research; A.P.S. and L.E. performed comparative genomics and molecular phylogenetic analyses; A.P.S. performed transcriptomic and proteomic; A.P.S. performed in situ hybridization; A.P.S. and B.G. performed histological analyses; A.P.S. and L.E. identified Hoxc13 binding motifs; M.C., I.B., W.D. and K.V. performed promoter activation assays; M.C., S.D. and K.V. generated and bred hoxc13 knockout frogs; M.C., M.B.C., I.B. and K.V. investigated the interaction of Hoxc13 with predicted binding sites; M.C., A.P.S., M.B.C., W.D., K.V. and L.E. analyzed data; and M.C., A.P.S., K.V. and L.E. wrote the first draft of the manuscript; all authors contributed to the revision of the manuscript; K.V. and L.E. supervised the study and share last-authorship.

## Competing interests
The authors declare no competing interests.
