## [Peer Review File · Nature Communications]

Evolutionary origin of Hoxc13-dependent skin appendages in amphibiansREVIEWER COMMENTS

Reviewer #1 (Remarks to the Author):

In this study, Carron et al. explore the evolutionary path to changes in the function of Hoxc13 in controlling the production of tetrapod cornified appendages. Hoxc13 was previously shown to control the development of limbs and the expression of keratin in mammalian hair and nail matrices through direct binding on promoters of hair keratin genes. Using phylogenetic analyses, authors show that keratin gene families exist in stem tetrapods and were lost in limbless amphibians. In addition, in *Xenopus tropicalis* frogs, orthologs of hair keratins (krt34 and krt59) are expressed in claws. From these observations, authors hypothesized that keratins of both amphibians and amniotes are regulated by Hoxc13. Using in situ hybridization in *X. tropicalis* and in vitro cell transfection, they demonstrate that Hoxc13 and krt34 co-localize in the epithelium of developing claws and that their expression is directly activated by Hoxc13. They generate Hoxc13 mutant frog individuals using CRISPR/Cas9-induced gene editing, and show that at least one copy of Hoxc13 allele is necessary for correct expression of krt34 and krt59 and subsequent claw development.

The manuscript is written in a straightforward, crystal-clear manner, and the study is sound, based on a complete set of developmental experiments adapted to the chosen model and an elegant and solid series of functional analyses. Overall, and with the exception of histological work (see below), I find the work robust, and its potential impact strong, as the function of Hox genes in late tissue formation and skin appendages is rarely explored. In addition, authors come up with an interesting evolutionary model in Hoxc13 primordially regulated the development of the tail fin in jawed vertebrates and has been co-opted in paired fins of lobe-finned fishes (i.e., the precursors of tetrapod limbs) to control keratin expression through novel binding sites, and in mammalian skin to control hair formation. In this model, and contrary to previous suggestions, claws of frogs and amniotes are homologous. The manuscript comprises a very interesting discussion on putative genetic and molecular events underlying mechanisms of evolutionary transitions in Hoxc13 function.

Major comments:

1. This work is certainly worth publishing provided authors can perform further histological work; for now, I find results of expression analyses and mutant phenotypes unconvincing.

I am indeed first concerned about the quality of histological work, which is central to the conclusions of the study. On several of the provided pictures, tissues are damaged (e.g. negative control for Hoxc13 in situ hybridization in Fig. 3) such that the claw is partially or largely removed. This is important, as authors histological observation to conclude that claws are missing in mutant individuals shown in Fig. 4 (but could they be rather than damaged by poor histological quality?). Images also appear to be provided at different developmental stages (in Fig. 3, the mesenchyme is thicker in krt34 in situs compared to Hoxc13 in situs, and the pigmentation of the claw is different). Please provide better histology pictures for Fig. 3 and 4, at homogeneous stages, as well as the number of biological replicates.

Second, in Fig. 4, the histology of the bones and epithelium looks very different between mutant and

control individuals, despite authors' claims that only claws are affected. Higher magnification pictures should be provided, and further immuno-histological work should be performed. Antibodies against keratins / keratinocyte markers likely exist, and histological stains / cell-level imaging resolution is feasible in *Xenopus* embryos.

2. I find it difficult to draw evolutionary conclusions at the entire phylogenetic scale of tetrapods by looking at tissue development in a single species. Functional experiments are understandably challenging in many species (and are performed in a very strong fashion in the present study, which is to be commended), but not expression analyses. The evolutionary scenario proposed by authors should be further supported by expression work in other groups (i.e. birds, reptiles), choosing species in which embryonic specimens can be easily obtained (e.g., chicken, quail, anole lizard).

Minor comments:

1. For coherence and to strengthen conclusions, it would be interesting to also explore the colocalization of Hoxc13 with krt59. Why was only one of the two targets tested?

2. The sentence "demonstrated that biallelic out-of-frame mutations of hoxc13 abolished the differentiation of toe tip keratinocytes into cornified claws" is an overstatement. The impairment of Hoxc13 may cause other defects indirectly modifying claw structure (e.g., absence of keratinocyte specification or proliferation, defect of epithelium formation, etc). Please rephrase and further discuss this point.

3. The sentence "Cornified skin appendages are characteristic features of important phylogenetic clades of terrestrial mammals, exemplified by hair in mammals and feathers in birds" is misleading (birds are not terrestrial mammals).

4. The sentence "hair keratins of both amphibians and amniotes are regulated by Hoxc13" is misleading (amphibians have no hair).

5. A few spelling mistakes here and there (e.g., title of Fig. 2)

Reviewer #2 (Remarks to the Author):

Keratins are major intermediate filament proteins and are grouped into type I and II in vertebrates. Their functions and expressions are significantly related to the diversification and evolution of appendages in terrestrial animals. In particular, those of hair/nail/claw keratin genes remain unknown largely and controversial. In this manuscript, the authors identified a large number of keratin sequences and discovered the origins of terrestrial hair/nail/claw keratin genes (krt34/59) which might be evidence for the evolution of skin appendages to adapt to terrestrial environment in animal evolution. Moreover, findings of hox binding sites on their upstream is a contribution for further understanding of keratin gene evolution in animals.

Although krt34/59 transcriptional regulation is the most important argument of this paper, this is somewhat lacking some important experiments about those of these keratin genes by hoxc13. However, overall, the manuscript is well-written and the conclusions are logical. Some issues for publication should be improved in the following comments.

Major comments:

To strengthen the authors' claims, they should show evidence of the direct regulation of krt34/59 by Hoxc13: for example, ChIP-Seq and Cut&Tag using anti-Hoxc13 antibody or disruption of these hox binding sites by using CRISPR-Cas9. These experiments are not very difficult in *X. tropicalis*, and at least one of them is necessary to solidify the value of this paper.

Minor comments:

1. Page 6 Line 179: "CRISPR/Cas9" should be changed to "CRISPR-Cas9".
2. Figure 3: Are krt34/59 and hoxc13 expressed in the same cell? It would be nice to have in situ hybridization data showing that hoxc13 directly regulates these keratins.
3. To claim that krt34/59 is the origin of hair keratin requires demonstrating that it differs in amino acid sequence and composition from other cytokeratins. It is also important to compare their biochemical characteristics based on the sequences of mammalian hair keratins.
4. HO (Digit I, II, III) may be affected by anterior regions in limb development. Shh is an important factor in amniote skin appendage formation, and it functions as a posterior factor (ZPA) in limb development. Any consideration of the relationship between Shh and the identity of these anterior digits and the expression of hair keratin would be of greater interest to developmental biologists.
5. Page 19: Figure 1e should be moved and integrated into Figure 5 for the reader to easily understand the co-evolution of hair keratin and hoxc13 regulation.
6. In mammals, it seems that Hoxc13 is involved in not only hair keratin but cytokeratin expression (e.g., Tkatchenko et al., 2001). In hoxc13 KO frogs, is there any effect on the expression of other keratin genes and the epidermis and epithelial organogenesis?
7. Do axolotls and other amphibians have unique claws? In addition, although Fig. 5 states that the ancestors of tetrapods had claws, it is necessary to provide the literature that supports this claim and some explanation.

Reviewer #3 (Remarks to the Author):

The manuscript by Carron, Sachslehner, et al proposes a new hypothesis on the evolution of skin appendages. The data is presented in a concise and clear manner and supports the conclusions of the authors. All experiments were designed and performed following the best practices in the field.

I only have a few minor comments presented below in the order that they appear in the paper, rather than in the order of importance:

1. Lines 69-72: It is necessary to expand more on the previous hypothesis to place this study better in the context
2. Line 113: It is necessary to indicate when the authors refer to human and frog genes, as it took me a while krt59 which is only visible in Figure 1
3. Figure 1: The resolution of the figure was very low but I assume that this is due to the submission process. The term 'synteny' could be used in the title of the figure. In Fig 1a and b, it is necessary to add a rectangular around the hair keratins to make it easier for the reader to identify them. Krt70 is in lower case. KRT80 and KRT8 have very similar colors. In the legend, please provide references for the dating of the nodes.
4. Figure 2: I didn't find a reference to Fig2b in the text.
5. Figures 2-4: Indicate the age of the animals in the legend.
6. Line 132: Again, it was difficult to find krt53
7. Section on the deletion of Hoxc13: Please add a supplementary figure/table with more information on the mutations generated
8. Discussion: Provide additional background on the previous assumption that claws evolved independently for the readers to better comprehend the novelty of the results presented here.
9. Discussion: Surprisingly there is no reference to beta keratins/proteins when discussing feather development. This part of the discussion needs to be better documented.
10. Tissue sampling and fixation: Given the harsh and long fixation process, it is surprising that the RNA is still intact. Were there any precautions taken for the preparation of samples for in situ hybridisations?
11. Histology: I didn't see any figures with Eosin staining.

Response to the reviewers

Reviewer #1 (Remarks to the Author):

In this study, Carron et al. explore the evolutionary path to changes in the function of Hoxc13 in controlling the production of tetrapod cornified appendages. Hoxc13 was previously shown to control the development of limbs and the expression of keratin in mammalian hair and nail matrices through direct binding on promoters of hair keratin genes. Using phylogenetic analyses, authors show that keratin gene families exist in stem tetrapods and were lost in limbless amphibians. In addition, in *Xenopus tropicalis* frogs, orthologs of hair keratins (krt34 and krt59) are expressed in claws. From these observations, authors hypothesized that keratins of both amphibians and amniotes are regulated by Hoxc13. Using in situ hybridization in *X. tropicalis* and in vitro cell transfection, they demonstrate that Hoxc13 and krt34 co-localize in the epithelium of developing claws and that their expression is directly activated by Hoxc13. They generate Hoxc13 mutant frog individuals using CRISPR/Cas9-induced gene editing, and show that at least one copy of Hoxc13 allele is necessary for correct expression of krt34 and krt59 and subsequent claw development.

The manuscript is written in a straightforward, crystal-clear manner, and the study is sound, based on a complete set of developmental experiments adapted to the chosen model and an elegant and solid series of functional analyses. Overall, and with the exception of histological work (see below), I find the work robust, and its potential impact strong, as the function of Hox genes in late tissue formation and skin appendages is rarely explored. In addition, authors come up with an interesting evolutionary model in Hoxc13 primordially regulated the development of the tail fin in jawed vertebrates and has been co-opted in paired fins of lobe-finned fishes (i.e., the precursors of tetrapod limbs) to control keratin expression through novel binding sites, and in mammalian skin to control hair formation. In this model, and contrary to previous suggestions, claws of frogs and amniotes are homologous. The manuscript comprises a very interesting discussion on putative genetic and molecular events underlying mechanisms of evolutionary transitions in Hoxc13 function.

Major comments:

1. This work is certainly worth publishing provided authors can perform further histological work; for now, I find results of expression analyses and mutant phenotypes unconvincing. I am indeed first concerned about the quality of histological work, which is central to the conclusions of the study. On several of the provided pictures, tissues are damaged (e.g. negative control for Hoxc13 in situ hybridization in Fig. 3) such that the claw is partially or largely removed.

Response:

We thank the reviewer for the positive general comments and for the constructive criticism of the histological images. As suggested by the reviewer, we have performed further histological work and present stainings of improved quality in Figure 3a-f and Figure 4f of the revised manuscript. Due to the material properties of the frog claws, it is difficult to obtain sections on which all parts are well preserved. As we have chosen samples from older frogs with larger claws, the pigmentation of the claws (brown) is stronger than the signals from the in situ hybridization (blue), which were nevertheless clear and well controlled by absence of signals in stainings with sense riboprobes.

This is important, as authors histological observation to conclude that claws are missing in mutant individuals shown in Fig. 4 (but could they be rather than damaged by poor histological quality?). Images also appear to be provided at different developmental stages (in Fig. 3, the mesenchyme is thicker in krt34 in situs compared to Hoxc13 in situs, and the pigmentation of the claw is different). Please provide better histology pictures for Fig. 3 and 4, at homogeneous stages, as well as the number of biological replicates.

Response:

As suggested by the reviewer, stainings of improved quality are shown in Figure 3a-f and Figure 4f. The number of biological replicates is shown in the legends.

Second, in Fig. 4, the histology of the bones and epithelium looks very different between mutant and control individuals, despite authors' claims that only claws are affected. Higher magnification pictures should be provided, and further immuno-histological work should be performed. Antibodies against keratins / keratinocyte markers likely exist, and histological stains / cell-level imaging resolution is feasible in *Xenopus* embryos.

Response:

In the revised manuscript, images of better quality are shown in Figure 4. We performed H&E and Masson trichrome-stainings and did not observe consistent differences between HI toes of KO and WT frogs except for the cornified claws in WT. However, the variability in the plane of sectioning and partial damage of the tissue during histological processes, does not allow us to exclude minor morphological differences between WT and *Hoxc13* KO toes. Therefore, we have deleted the statement "which showed no other morphological abnormalities of toes" in the description of HI toes of KO frogs (Fig. 4g).

We do not have antibodies specifically binding to *Xenopus* keratins and therefore, we have not performed immunohistological stainings. To determine keratin expression, we analyzed the transcriptome of the toes (Supplementary Figure 3). The results of this analysis confirmed the decreased expression of hair keratin homologs and showed upregulation of *krt78.3* and *krt78.4*.

2. I find it difficult to draw evolutionary conclusions at the entire phylogenetic scale of tetrapods by looking at tissue development in a single species. Functional experiments are understandably challenging in many species (and are performed in a very strong fashion in the present study, which is to be commended), but not expression analyses. The evolutionary scenario proposed by authors should be further supported by expression work in other groups (i.e. birds, reptiles), choosing species in which embryonic specimens can be easily obtained (e.g., chicken, quail, anole lizard).

Response:

As suggested by the reviewer, we have performed gene expression studies in other groups, namely the axolotl and the green anole lizard. The axolotl represents salamanders as the second main clade of amphibians (besides anurans and in contrast to caecilians) that has toes (Figure 5 of the revised manuscript). The green anole lizard was the reptilian model species in which expression of hair keratin homologs in claws was demonstrated for the first time outside of the clade of mammals (Eckhart et al. PNAS 2008). The new results are shown in Figure 5 (axolotl) and Supplementary Figure S2 (anole lizard) of the revised manuscript. In both species, hair keratin homologs and *Hoxc13* are co-expressed in toes but not the skin of other parts of the body. As in the other parts of our study, we did not investigate gene expression during embryonic development but in adult tissues. In situ hybridization of *krt34*, a hair keratin homolog, in the axolotl demonstrates expression in the epithelium of the toe tips which is covered by a superficial layer of keratinocytes reminiscent of the *Xenopus* claws, as can be seen in albino animals. The association of these claw-like structures with expression of hair keratin homologs and *Hoxc13* strongly supports the hypothesis on the evolutionary origin of *Hoxc13*-dependent skin appendages in amphibians (Fig. 6).

The expression of *Hoxc13* in toe tips of chicken embryos was reported previously (Reference 25). However, birds are not a suitable model for the study of the interactions between *Hoxc13* and ancestral keratins in claws, because hair keratin homologs have been substituted by sauropsid-specific cysteine-rich keratins (Eckhart et al. 2008; Ehrlich et al. 2020) during the evolution of birds.

Minor comments:

1. For coherence and to strengthen conclusions, it would be interesting to also explore the colocalization of Hoxc13 with *krt59*. Why was only one of the two targets tested?

Response:

As suggested by the reviewer, we have performed *krt59* in situ hybridization. In contrast to previous experiments, in which the morphology of the claw was incompletely preserved, new staining gave images of good quality, like the one shown in Figure 3c of the revised manuscript.

2. The sentence “demonstrated that biallelic out-of-frame mutations of *hoxc13* abolished the differentiation of toe tip keratinocytes into cornified claws” is an overstatement. The impairment of Hoxc13 may cause other defects indirectly modifying claw structure (e.g., absence of keratinocyte specification or proliferation, defect of epithelium formation, etc). Please rephrase and further discuss this point.

Response:

As suggested by the reviewer, we have changed this sentence which now reads:

“These phenotyping data demonstrated that biallelic out-of-frame mutations of *hoxc13* abolished, either directly or indirectly, the differentiation of toe tip keratinocytes into cornified claws whereas single alleles of wildtype or in-frame-mutant *hoxc13* sufficed to induce claw formation.”

3. The sentence “Cornified skin appendages are characteristic features of important phylogenetic clades of terrestrial mammals, exemplified by hair in mammals and feathers in birds” is misleading (birds are not terrestrial mammals).

Response:

We are sorry for this mistake in writing. “mammals” was changed to “vertebrates” in the revised manuscript.

4. The sentence “hair keratins of both amphibians and amniotes are regulated by Hoxc13” is misleading (amphibians have no hair).

Response:

To be more precise, we have changed “hair keratins” to “hair keratin homologs” in the revised manuscript.

5. A few spelling mistakes here and there (e.g., title of Fig. 2)

Response:

We thank the reviewer for pointing to this typo in the title of Fig. 2. “kertain” was changed to “keratin” in the revised manuscript. We have also carefully corrected errors in the rest of the manuscript.

Reviewer #2 (Remarks to the Author):

Keratins are major intermediate filament proteins and are grouped into type I and II in vertebrates. Their functions and expressions are significantly related to the diversification and evolution of appendages in terrestrial animals. In particular, those of hair/nail/claw keratin genes remain unknown largely and controversial. In this manuscript, the authors identified a large number of keratin sequences and discovered the origins of terrestrial hair/nail/claw keratin genes (*krt34/59*)

which might be evidence for the evolution of skin appendages to adapt to terrestrial environment in animal evolution. Moreover, findings of hox binding sites on their upstream is a contribution for further understanding of keratin gene evolution in animals. Although *krt34/59* transcriptional regulation is the most important argument of this paper, this is somewhat lacking some important experiments about those of these keratin genes by *hoxc13*. However, overall, the manuscript is well-written and the conclusions are logical. Some issues for publication should be improved in the following comments.

Major comments:

To strengthen the authors' claims, they should show evidence of the direct regulation of *krt34/59* by *Hoxc13*: for example, ChIP-Seq and Cut&Tag using anti-*Hoxc13* antibody or disruption of these hox binding sites by using CRISPR-Cas9. These experiments are not very difficult in *X. tropicalis*, and at least one of them is necessary to solidify the value of this paper.

Response:

We appreciate the comments and suggestions of the reviewer regarding this important point. We would like to respond in two parts:

First, we want to emphasize that evidence for the direct regulation of *krt34* and *krt59* by *Hoxc13* is provided in Figure 3h and i. In these experiments, we have co-transfected cells with a *Hoxc13* expression vector and a luciferase reporter under the control of the *krt34* or *krt59* proximal promoter in which the predicted *Hoxc13* binding sites were present either with their wildtype sequence or with mutations that abolish binding of *Hoxc13*. The results shown in the figure demonstrate that *Hoxc13* activates transcription from both *krt34* or *krt59* promoters. Mutation of the consensus *Hoxc13* binding sites significantly reduced transcriptional activity. Importantly, the homologous sites in the promoters of human hair keratins were shown to be sites of direct binding of *Hoxc13* (Jave-Suarez et al. 2000), as we mention in the manuscript.

Second, we have considered additional experiments involving the disruption of hox binding sites by using CRISPR-Cas9, ChIP-Seq and Cut&Tag using anti-*Hoxc13* antibody, as suggested by the reviewer. These experiments were not feasible because of technical issues described in the next paragraph. However, we performed a ChIP experiment in cultured frog cells expressing tagged *Hoxc13*, and the results support the hypothesis. We show the new data in a Figure for the reviewer (attached below), which will be published as part of the Peer review file linked to the manuscript.

The specific disruption of the *Hoxc13* binding sites by using CRISPR-Cas9 was not possible because the promoters of *krt34* and *krt59* do not contain suitable protospacer adjacent motif (PAM) sequences that would allow the specific targeting of these sites. We do evidently agree that a ChIP or Cut&Tag experiments in frog tissue would help to further test the hypothesis that *krt34* and *krt59* genes are directly regulated by *Hoxc13*. However, to investigate a direct interaction in the toe tips of *Xenopus* (where *Hoxc13* and the keratin genes are expressed) antibodies recognizing *Xenopus* *Hoxc13* would be required. We have scanned the datasheets of a wide range of commercially available *Hoxc13* antibodies. Either information on the immunogen was not sufficiently provided or the possible epitopes were not or poorly conserved in *Xenopus*. When conservation of the immunogen was relatively high, the antibody was directed to the DNA binding domain, which would evidently be incompatible with its use for a ChIP or Cut&Tag experiment.

As a best possible proxy, we have generated a FLAG-tagged version of *Xenopus tropicalis* *Hoxc13* to transfect *Xenopus* XTC-2 cells (Pudney M, Varma MG, Leake CJ. Establishment of a cell line (XTC-2) from the South African clawed toad, *Xenopus laevis*. *Experientia*. 1973;29:466-7. doi: 10.1007/BF01926785), performed ChIP with an antibody against the FLAG-tag and quantified

precipitated promoter segments of *X. laevis krt34* and *krt59* genes containing the Hoxc13 binding motifs that are shown in Figure 3g. Appropriate control experiments such as the analysis of non-transfected cells, transfection with non-tagged Hoxc13, no antibody controls for IP and quantification of DNA segments lacking Hoxc13 binding sites were performed in parallel. This experimental system has limitations. We used the XTC-2 cell line because it was the only frog cell line available that has been used successfully in previous transfection experiments. As a fibroblast-like cell line, XTC-2 cells do not normally express the *krt34* and *krt59* genes, and hence it is not sure whether their promoter regions are accessible. We reasoned that the high levels of ectopic Hoxc13 expression could possibly allow interactions of Hoxc13 with the endogenous *krt34* and *krt59* promoters. XTC-2 cells are derived from *Xenopus laevis* and hence have an allotetraploid genome that contains the *krt34.L/krt34.S* and *krt59.L/krt59.S* homeologous alleles.

The results of our ChIP experiment are summarized in the Figure for the reviewer. The promoter regions of *krt34.L*, *krt34.S* and *krt59* (at least one of the homeolog) are clearly enriched in the lysate derived from the cells transfected with FLAG-tagged Hoxc13. By contrast, control gene segments of *tnfa* and *eef1a1* were not enriched by CHIP with FLAG-tagged Hoxc13. These results obtained in *X. laevis* cells support the hypothesis that Hoxc13 directly interacts with the promoter regions of *Xenopus krt34* and *krt59* genes.

Figure for the reviewer: ChIP analysis of Hoxc13 interactions with the promoters of *krt34* and *krt59*. *Xenopus laevis* XTC-2 cells were transfected with *Xenopus tropicalis* Hoxc13 that was either untagged or FLAG-tagged. Non-transfected cells were analyzed for control. ChIP was performed with anti-FLAG antibody. No antibody was used in control experiments. Quantitative PCRs were performed with primers specific for the promoter regions of *krt34.S*, *krt34.L* and *krt59* as well as primers for *tnfa.L* and *eef1a1.S* as controls. Fold enrichment values were calculated as the ratio between ChIP-derived DNA versus input DNA. Bars and error bars indicate means and standard deviations, respectively, of three technical replicates. Differences were assessed by one-way ANOVA, followed by Tukey's multiple comparison test. *p<0.05, **p<0.01, ***p<0.001.

Methods for the Figure for the reviewer

Construction of pcDNA3.1_3xFlag_XtHoxc13

For the generation of the Flag-tagged Hoxc13, the *X. tropicalis hoxc13* coding sequence was inserted into the pcDNA3.1_3xFlag_IFIT2 plasmid (<https://www.addgene.org/53555/>), where the IFIT2 sequence was substituted for Hoxc13 upon *Bam*HI and *Xba*I digestion, using the Genbuilder® cloning kit (Genscript) according to manufacturer's instructions. The sequences of the constructs

were verified by Sanger sequencing after amplification with the primers 5'-ACGTAATACGACTCACTATAG-3' and 5'-AGTAGTGCCAGGCATCAGC-3'.

Transient transfection for chromatin immunoprecipitation (ChIP)

XTC-2 cells were cultured in 0.66x complete medium, where 1x medium is Dulbecco's Modified Eagle Medium (DMEM, Gibco, Waltham, MA, USA), supplemented with 10% fetal calf serum. Cells were seeded in standard tissue culture dishes (58 cm²) and transfected with either pcDNA3.1_3xFlag_XtHoxc13 or with pCS2+_XtHoxc13 plasmid (11 µg/dish) or not transfected (no transfection control), using lipofectamine 3000 (high) following the manufacturer's instructions.

Crosslinking and shearing

One day after transfection XTC-2 cells were washed with amphibian PBS (0.66xPBS) before crosslinking with 1% formaldehyde for 10 min. The crosslinking reaction was stopped by 5 min incubation in 0.125 M glycine/PBS on a rocker. Cells were washed with ice-cold PBS and collected via scraping in 8 ml PBS. Cells were pelleted for 5 min at 4°C and 1000 g. Supernatant was discarded and the cell pellet was dissolved in 300 µL RIPA buffer and incubated for 10 min on ice. The lysate was added to 1.5 ml Bioruptor® Plus TPX microtubes (Diagenode), and DNA was sheared using the Bioruptor® Plus (Diagenode) with 1 run of 10 cycles [30 sec "ON", 30 sec "OFF"]. After centrifugation for 10 min at 4°C and 8000 g, the supernatant was recovered and aliquots of 50 µL were stored at -80°C. An aliquot was taken for determining the nucleic acid and protein concentrations.

Immunoprecipitation

Samples were thawed, adjusted to equal nucleic acid concentration in RIPA buffer and further diluted to a final volume of 600 µL. A 10 µL sample was taken for later use as input sample. Samples were cleared by centrifugation and incubated overnight at 4°C with anti-FLAG® M2 agarose beads (Sigma Aldrich, A2220). Lysate from Flag-Hoxc13 transfected cells was incubated with agarose beads as a no-antibody control. Immunocomplexes were precipitated by centrifugation, washed extensively and finally eluted according to a published protocol (Blythe SA, Reid CD, Kessler DS, Klein PS. Chromatin immunoprecipitation in early *Xenopus laevis* embryos. Dev Dyn. 2009;238:1422-32. doi: 10.1002/dvdy.21931). After reversal of the crosslinks and proteins digestion, DNA purified from the samples was RNase treated and re-purified using a QIAquick PCR Purification Kit (Qiagen11).

PCR

DNA derived from the ChIP and input samples was used for quantitative real time PCR with primers specific for the promoter regions of *krt34.S*, *krt34.L*, *krt59* (primers bind to both the L and S homeologs) and primers for *eef1a1.S* and *tnfa.L* as negative controls. For each setup, a ratio between the fitted values derived from the extrapolated Ct measurements of the amplicon from the ChIP-derived DNA versus the input DNA was calculated. To validate the data distribution, the Shapiro-Wilk normality test was applied to the triplicate measurements for each gene. Subsequently, one-way ANOVA was performed individually for *krt34.L*, *krt34.S*, *krt59*, *tnfa.L* and *eef1a1.S*, followed by Tukey's multiple comparison test.

Table. PCR primers

Gene	Forward primer	Reverse primer
krt34.L	GCCTTTGTGTGCAGATTCCA	CAGCTTTGGGTGTTGACAGT
krt34.S	CCTTAGTGAGCAGATTCCAAAAGA	CGTCAGTCTTGCTGGTACG
krt59	TGCCAGAATGCGTAAATCTTCA	CCTAAAGTGCTGGCAATGCA
tnfa.L	GCTCAAGGATAACTCCATCG	AACCAAGTGGCACCTGAATG
eef1a1.	CCTGAATCACCCAGGCCAGATTGGT	GAGGGTAGTGTGAGAAGCTCTCCAC

Minor comments:

1. Page 6 Line 179: “CRISPR/Cas9” should be changed to “CRISPR-Cas9”.

Response:

As suggested by the reviewer, we have changed “CRISPR/Cas9” to “CRISPR-Cas9”.

2. Figure 3: Are *krt34/59* and *hoxc13* expressed in the same cell? It would be nice to have in situ hybridization data showing that *hoxc13* directly regulates these keratins.

Response:

We provide new images of in situ hybridizations in the revised Figure 3. *krt34*, *krt59* and *hoxc13* are expressed in the same cells.

3. To claim that *krt34/59* is the origin of hair keratin requires demonstrating that it differs in amino acid sequence and composition from other cytokeratins. It is also important to compare their biochemical characteristics based on the sequences of mammalian hair keratins.

Response:

Our conclusion that *krt34* and *krt59* are orthologs of hair keratins is based on molecular phylogenetics (Figure 1c, d) that was calculated on the basis of amino acid sequence alignments. From this orthology, we infer that *krt34* and *krt59* and hair keratins have evolved from ancestral genes which were the evolutionary predecessors of hair keratins. Other reports, that are cited in our manuscript, have also concluded that the evolution of the type 1 and type 2 hair keratin families started in amphibians, however, without demonstrating expression of hair keratin homologs in amphibian claws or toes of *axlotl* and without showing their loss in legless amphibians (caecilians). In the Introduction, we write “Further studies led to the identification of hair keratin-like proteins, though distinguished from human hair keratins by a low content of cysteine, in amphibians [7,8,9].” With this statement, we have indicated that one of the characteristic features of hair keratins, i.e. the high content of cysteine, is not found in the orthologs in amphibians. In line with the comment of the reviewer, we further discuss this difference in the Discussion section, third paragraph: “While the regulation of hair keratin homologs by *Hoxc13* appears to have evolved in stem tetrapods, the high content of cysteine residues in hair keratins [34,6] is not shared by their orthologs in amphibians [9], suggesting that the initial evolution of keratin gene regulation preceded the evolution of cysteine-dependent crosslinking of hair keratin homologs.”

4. HO (Digit I, II, III) may be affected by anterior regions in limb development. *Shh* is an important factor in amniote skin appendage formation, and it functions as a posterior factor (ZPA) in limb development. Any consideration of the relationship between *Shh* and the identity of these anterior digits and the expression of hair keratin would be of greater interest to developmental biologists.

Response:

We thank the reviewer for raising this interesting point. We have performed RNA-seq analysis of WT (n=3) and *hoxc13* KO (n=3) toes. In contrast to the marked increases of *krt34* and *krt59* levels in KO samples (Supplementary Figure 3 of the revised manuscript), the mRNA levels of *Shh* signaling genes were not significantly altered by the knockout of *hoxc13*:

Table. Expression levels of *Shh* signaling genes in HI toes, determined by RNA-seq analysis

Gene	GenBank accession nr.	WT (mean \pm SD)	KO (mean \pm SD)	P-value (t-test)
shh	XM_031903906.1	10 \pm 8	12 \pm 6	0.791
ptch1	XM_031890959.1	177 \pm 22	217 \pm 6	0.109

gli1	XM_002939425.5	74 ± 30	80 ± 13	0.833
gli2	XM_002943283.5	103 ± 21	122 ± 52	0.672
gli3	XM_012965492.3	106 ± 3	85 ± 63	0.695

However, the Shh signaling is particularly important because inhibition of this signaling was previously reported to affect the development of claws in *X. tropicalis* and an indirect link to Hoxc13-dependent regulation of hair keratin homologs in adult claws is possible. Therefore, we discuss this topic and include relevant references in the Discussion section of the revised manuscript:

“For example, sonic hedgehog (Shh) is implicated in the evolution of skin appendages and plays a critical role in the morphogenesis of hair follicles [30,31,32]. Interestingly, the inhibition of Shh signaling in late stages of limb development allows the formation of claws on digits IV and V (HO toes) of *Xenopus tropicalis* [33]. The effects of Shh signaling on the expression of hair and claw keratins are likely indirect, whereas experimental data obtained in cultured cells indicate that Hoxc13 directly regulates the expression of human hair keratins [14] and *Xenopus* claw keratins (Fig. 3h, i).”

5. Page 19: Figure 1e should be moved and integrated into Figure 5 for the reader to easily understand the co-evolution of hair keratin and hoxc13 regulation.

Response:

We thank the reviewer for this suggestion. We have considered combining the Figures 1e and 5 (which is figure 6 in the revised manuscript), but we decided to keep them separate for the following reasons. Figure 1e shows the co-evolution of hair keratin homologs and limbs/digits, but does not contain information about Hoxc13. The focus of this figure is on the loss of hair keratin homologs in legless caecilians. Figure 5 shows the link between the evolution of Hoxc13 and the evolution of hair keratins, with Hoxc13 being a regulator of hair keratin expression. The inclusion of caecilians in Figure 5 would not help to test this hypothesis because hair keratin homologs are not expressed in caecilians whereas Hoxc13 is retained, suggesting hair keratin-independent and not tetrapod-specific functions of Hoxc13 in development.

6. In mammals, it seems that Hoxc13 is involved in not only hair keratin but cytokeratin expression (e.g., Tkatchenko et al., 2001). In hoxc13 KO frogs, is there any effect on the expression of other keratin genes and the epidermis and epithelial organogenesis?

Response:

Tkatchenko et al., 2001 have shown that transgenic overexpression of Hoxc13 in the skin of mice leads to aberrant expression of several genes (mainly keratin-associated proteins, KRTAPs, that are normally expressed in the hair shaft) and disturbance of hair follicle organization. The data from the study in the mouse do not suggest candidate genes of Hoxc13 targets in *Xenopus*. Therefore, we performed a genome-wide gene expression study in WT and hoxc13 KO toes. RNA-seq confirmed the downregulation of hair keratin homologs *krt34* and *krt59* in HI toes of KO frogs, whereas epithelial keratins *krt78.3* and *krt78.4* were upregulated. These data are shown in Supplementary Figure 3 of the revised manuscript.

Currently, we do not have evidence for effects of hoxc13 knockout on the epidermis and epithelial organogenesis except for claw formation, however, this conclusion is preliminary because the number of knockout frogs was not sufficient for supporting potential quantitative differences in epithelial parameters. In follow-up studies we will investigate the effect of Hoxc13 on specialized epithelial structures such as nuptial pads (Maddin et al. 2009) developing under hormonal control on forelimbs of male *Xenopus* frogs.

7. Do axolotls and other amphibians have unique claws? In addition, although Fig. 5 states that the ancestors of tetrapods had claws, it is necessary to provide the literature that supports this claim and some explanation.

Response:

We thank the reviewer for raising this important point which was not addressed in a clear way in the original manuscript. In the revised manuscript we provide data on claw-like structures that are present in the axolotl. Importantly, we also demonstrate that Hoxc13 and hair keratin homologs are expressed in the toes containing these claw-like structures. To the best of our knowledge, this is the first histological demonstration of claws in the axolotl. The conservation of hair keratin expression supports our hypothesis. The new data are shown in Figure 5 of the revised manuscript.

Our data suggest that the hair keratin homologs were expressed in claws, toe pads or homologous structures of the last common ancestor of tetrapods. However, to the best of our knowledge, there is no fossil evidence for the presence of claws in stem tetrapods. In accordance with the available evidence, we propose that “Hoxc13-dependent expression of hair keratin homologs evolved already in stem tetrapods, presumably as a mechanism for protecting toe tips”, as it is stated in the Abstract.

Reviewer #3 (Remarks to the Author):

The manuscript by Carron, Sachslehner, et al proposes a new hypothesis on the evolution of skin appendages. The data is presented in a concise and clear manner and supports the conclusions of the authors. All experiments were designed and performed following the best practices in the field.

I only have a few minor comments presented below in the order that they appear in the paper, rather than in the order of importance:

1. Lines 69-72: It is necessary to expand more on the previous hypothesis to place this study better in the context

Response:

We thank for the reviewer for the helpful comments. As suggested by the reviewer, we provide more information on the previous hypothesis in the revised manuscript.

2. Line 113: It is necessary to indicate when the authors refer to human and frog genes, as it took me a while krt59 which is only visible in Figure 1

Response:

Information on the species was added to the names of keratins, as suggested by the reviewer.

3. Figure 1: The resolution of the figure was very low but I assume that this is due to the submission process. The term ‘synteny’ could be used in the title of the figure. In Fig 1a and b, it is necessary to add a rectangular around the hair keratins to make it easier for the reader to identify them. Krt70 is in lower case. KRT80 and KRT8 have very similar colors. In the legend, please provide references for the dating of the nodes.

Response:

We thank the reviewer for these comments and suggestions which have helped us to improve the figure and the legend. The resolution of the figure was indeed low because of the file conversion during the initial submission. A high quality figure is provided now. Synteny is used in the revised title of the figure. Rectangulars mark hair keratins. In line with GenBank guidelines, we use upper case for

genes of human and lungfish and lowercase for amphibians. Colors were adapted. A reference for dating the nodes is provided in the legend.

4. Figure 2: I didn't find a reference to Fig2b in the text.

Response:

We have added a reference to Fig. 2b in the text.

5. Figures 2-4: Indicate the age of the animals in the legend.

Response:

The age of the animals is indicated in the revised legends.

6. Line 132: Again, it was difficult to find *krt53*

Response:

We have specified that we refer to *X. tropicalis krt53*.

7. Section on the deletion of *Hoxc13*: Please add a supplementary figure/table with more information on the mutations generated

Response:

As suggested by the reviewer, we have added the new supplementary figures 1 and 2, in which more information on the mutations is provided.

8. Discussion: Provide additional background on the previous assumption that claws evolved independently for the readers to better comprehend the novelty of the results presented here.

Response:

Additional details on the previous hypothesis of claw evolution was added.

9. Discussion: Surprisingly there is no reference to beta keratins/proteins when discussing feather development. This part of the discussion needs to be better documented.

Response:

We have added references to beta-keratins and other specific feather proteins in the Discussion.

10. Tissue sampling and fixation: Given the harsh and long fixation process, it is surprising that the RNA is still intact. Were there any precautions taken for the preparation of samples for in situ hybridisations?

Response:

We thank the reviewer for this comment which help to us notice that important information on the pH was missing in the Methods. In the revised manuscript we have added the information that decalcification was performed at pH 8.0 which helps to preserve the RNA. Although the fixation of the tissue may decrease the subsequent detection of mRNA to some extent, the mRNA in situ hybridization protocol yields reliable signals.

11. Histology: I didn't see any figures with Eosin staining.

Response:

We provide hematoxylin and eosin stainings of improved quality in revised Figure 4f, g and in the new Figure 5c.

Other changes made during the revision:

- The drawing of mosaic mutant frogs in Figure 4a was modified to indicate that some of the mosaic mutants lack cornified claws.
- The labeling of fishes in the left part of the schematic was removed to focus the schematic on the evolution of the Hoxc13 – keratin regulation in tetrapods.
- Typos were corrected.

REVIEWERS' COMMENTS

Reviewer #1 (Remarks to the Author):

The authors have done a very nice job at addressing all my comments, in particular by adding data in the axolotl, presented in a novel Figure (Figure 5). I have no further comments.

Reviewer #2 (Remarks to the Author):

The authors have addressed all concerns as detailed in the first round of review, and added extra data to support the conclusions. As a minor detail, I think the previous version of the field data (Figures 3a-f) looked better.

Reviewer #3 (Remarks to the Author):

The authors properly address all reviewers questions. No further comments.

Response to the reviewers' comments

Reviewer #1 (Remarks to the Author):

The authors have done a very nice job at addressing all my comments, in particular by adding data in the axolotl, presented in a novel Figure (Figure 5). I have no further comments.

Response:

We thank the reviewer for the positive comments.

Reviewer #2 (Remarks to the Author):

The authors have addressed all concerns as detailed in the first round of review, and added extra data to support the conclusions. As a minor detail, I think the previous version of the field data (Figures 3a-f) looked better.

Response:

We thank the reviewer for the positive comments. In the revised manuscript, we provide the original set of histological pictures (Fig. 3) as Supplementary Figure 1, so that both versions are accessible to the readers.

Reviewer #3 (Remarks to the Author):

The authors properly address all reviewers questions. No further comments.

Response:

We thank the reviewer for the positive comments.